# Loss-Aligned Structured Pruning for Large Language Models

## Abstract

Recent advances in large language models (LLMs) have achieved remarkable performance across diverse tasks, yet their increasing size poses significant storage and computational challenges. Model compression, particularly pruning, has emerged as a crucial strategy to reduce memory footprint and computation while preserving model performance. In this work, we present LASP, a Loss-Aligned Structured Pruning method that evaluates the contribution of individual model units, such as neurons and attention heads, to the overall performance, subsequently removing those deemed to be of low importance. By combining the activation magnitudes of model units with their gradients with respect to the loss, LASP defines an importance metric that is directly aligned with the model's objective, thereby ensuring the preservation of performance. To mitigate uncertainty caused by the limited calibration dataset used for importance estimation, LASP incorporates the Upper Confidence Bound (UCB) strategy, refining the selection of low-importance units. In implementation, LASP leverages a moving average to maintain running statistics and reduce storage overhead. Empirical results across diverse LLMs and benchmarks demonstrate that LASP outperforms state-of-the-art baselines, effectively balancing efficiency and performance, thus enabling the practical deployment of LLMs.

## 1 Introduction

Recent advances in large language models (LLMs) (Touvron et al., 2023a;b; OpenAI, 2023) have demonstrated remarkable capabilities in language understanding, reasoning, and problem-solving. With increasing model size, their performance continues to improve (Kaplan et al., 2020), highlighting the benefits of scaling. However, this rapid growth in parameter count also leads to substantial demands on storage and computational resources. As a result, reducing the memory footprint and computational cost of LLMs has become a central research focus. A variety of approaches have been explored to compress LLMs, such as pruning (Cheng et al., 2023), quantization (Gholami et al., 2021), and knowledge distillation (Goulami et al., 2021).

In this work, we focus on pruning, a widely used model compression technique that removes internal redundancies in neural networks while preserving model performance, ideally without requiring costly recovery fine-tuning. Pruning reduces the number of parameters and computational operations, which can significantly lower memory footprint and inference latency, making it particularly important for deploying LLMs in resource-constrained environments.

Existing pruning approaches for LLMs can be broadly categorized into unstructured and structured methods. Unstructured pruning (Sun et al., 2024), illustrated in Fig. 1a, sparsifies weight matrices by zeroing individual elements, resulting in generally sparse parameter matrices. Variants such as 2:4 or 4:8 sparsity patterns are also adopted to exploit GPU sparse acceleration units for faster inference. In contrast, structured pruning (Ma et al., 2023; Ashkboos et al., 2024; An et al., 2024), shown in Fig. 1b and Fig. 1c, removes entire rows, columns, or higher-level structures such as attention heads. By pruning contiguous blocks, structured methods produce models that are naturally aligned with hardware accelerators, thereby reducing both memory footprint and computational cost while preserving architectural compatibility.

However, existing pruning techniques face several limitations. Unstructured pruning can yield highly sparse weight matrices, yet it fails to deliver actual storage reduction because the zeroed-out

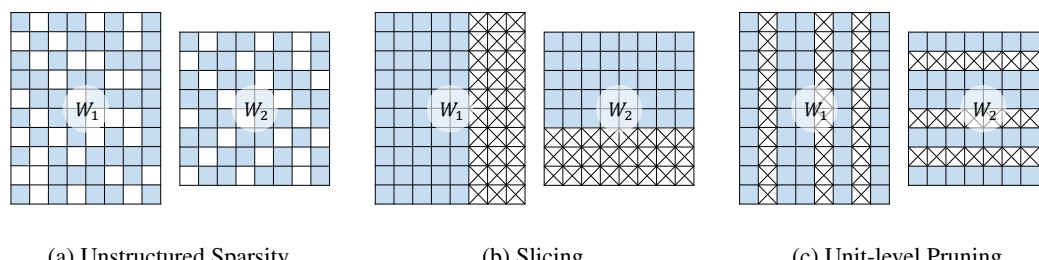

(a) Unstructured Sparsity          (b) Slicing          (c) Unit-level Pruning

Figure 1: Comparison of representative pruning methods applied to two consecutive weight matrices, $W_1$ and $W_2$, within a neural network. White grids indicate pruned parameters. **Unstructured sparsity** removes individual weights, producing irregular sparsity patterns, but does not reduce storage requirements. **Slicing** removes contiguous blocks of weights, partially reducing the matrix dimensions. **Unit-level pruning** removes entire units, corresponding to full columns in $W_1$ and full rows in $W_2$. Both slicing and neuron-level pruning effectively reduce storage requirements.

parameters must still be stored within the original dense tensor format. Given the critical demand for a low memory footprint in practical deployment, we steer clear of this approach and instead focus on structured pruning, which is inherently more hardware-friendly. Nevertheless, current structured pruning methods have their own drawbacks. Many gradient-based approaches, though hardware-compatible, often rely on computationally expensive operations such as estimating second-order Hessian information (Ma et al., 2023). Meanwhile, other prominent methods prune based on weight matrices, typically seeking to reconstruct them via low-rank decomposition (Ashkboos et al., 2024). While effective in practice, these methods primarily target the reconstruction of intermediate weight matrices, lacking an explicit alignment with the model's optimization objective, making it difficult to guarantee performance preservation.

Addressing the aforementioned challenges, we introduce LASP, a loss-aligned structured pruning method. Unlike existing approaches that primarily assess the importance of individual parameters, LASP evaluates the significance of model units, such as neurons and attention heads, by combining their activation magnitudes with gradients with respect to the model performance, thus defining a unit-level pruning metric directly aligned with the optimization objective. Units deemed unimportant are pruned together with all associated parameters, enabling the effective removal of redundancy while preserving model performance. To mitigate uncertainty arising from the limited calibration dataset used to estimate activations and gradients, LASP incorporates the Upper Confidence Bound (UCB) strategy (Auer et al., 2002), ensuring that low-importance units are pruned with high confidence. Moreover, in practical implementation, LASP leverages a simple moving average to maintain running statistics, thereby reducing storage overhead during pruning.

Extensive experiments across diverse LLMs and benchmarks demonstrate the effectiveness of our approach. At a 20% pruning ratio, the pruned models achieve lower loss on the calibration dataset than the original models, indicating that LASP effectively aligns pruning decisions with the optimization objective. Across multiple benchmarks, the pruned models maintain strong performance, preserving 93.5% of the original capability at a 25% pruning ratio and 90% at a 30% pruning ratio, while outperforming the state-of-the-art baseline.

In summary, our contributions are threefold:

- **Loss-Aligned Importance Metric**. LASP introduces an innovative metric that evaluates the importance of model units, allowing for the pruning of parameters associated with low-importance units. It ensures that pruning decisions are aligned with the model's objective.

- **Uncertainty-Aware Pruning**. LASP employs the UCB strategy to handle uncertainty in the limited calibration dataset, ensuring high-confidence selection of low-importance units.

- **Comprehensive Experimental Validation**. We conduct extensive experiments across diverse LLMs, i.e., the Llama (Touvron et al., 2023a), Llama2 (Touvron et al., 2023b), and Vicuna-v1.5 (Zheng et al., 2023) model series, to validate LASP. Our results show that LASP can efficiently compress LLMs while effectively preserving their performance.

## 2    RELATED WORK

**Pruning with Limited Data**. A recent line of research, closely related to our approach, focuses on the challenging task of pruning with limited data (Hubara et al., 2021; Frantar et al., 2022; Frantar & Alistarh, 2022; Kwon et al., 2022). These methods are highly desirable because they don't require any changes to the original training process and eliminate the need for computationally expensive retraining on the full dataset. To achieve this, their primary aim is to preserve model performance by leveraging a small amount of data, often called calibration data. They do this by solving a layer-wise reconstruction problem (Hubara et al., 2021) to mitigate the inevitable accuracy drop, which aims to minimize the change in a layer's output with respect to the calibration data. However, a key limitation of existing solvers is their reliance on the computationally heavy calculation of second-order Hessian inverses (Singh & Alistarh, 2020; Frantar et al., 2022), which makes them impractical and difficult to scale to the large hidden state sizes of modern Large Language Models (LLMs). The SparseGPT method (Frantar & Alistarh, 2023) offers a solution by developing a more efficient weight update procedure for LLMs, which uses synchronized second-order Hessian updates to circumvent this computational bottleneck.

**Structured Pruning**. Structured pruning approaches aim to identify and remove less important neurons or components while maintaining model performance. SliceGPT (Ashkboos et al., 2024) leverages PCA on weight matrices to selects the most informative subspaces for pruning and minimize reconstruction error. LLM-Pruner (Ma et al., 2023), on the other hand, evaluates the importance of each parameter using gradient and second-order information, aggregates these scores at the channel or neuron level, and prunes accordingly to enable efficient deployment of large models. Recent works have also explored integrating parameter-efficient finetuning with structured pruning, such as Compresso(Guo et al., 2023) and LoRAPrune (Zhang et al., 2024). These methods perform pruning during or after LoRA-style adaptation, leveraging the low-rank update signals to guide the removal of structural units.

## 3    METHODOLOGY

In this section, we first formulate the problem, then introduce the loss-aligned importance metric and the uncertainty-aware unit selection strategy. Afterwards, we outline the implementation of LASP.

### 3.1    PROBLEM FORMULATION

Given a pre-trained LLM $f$, a target pruning ratio $\gamma$, and a calibration dataset $\mathcal{D}_{\mathrm{cal}}$ containing sequences $x$, our goal is to conduct structured pruning by identifying a subset of units $\mathcal{U}_{\mathrm{prune}} \subset \mathcal{U}$ whose removal minimizes the degradation in model performance. Model performance is measured by a loss function $\mathcal{L}_{\mathrm{pred}}$, and the pruning objective is formally expressed as:

$$\mathcal{U}^*_{\mathrm{prune}} = \arg \min_{\mathcal{U}_{\mathrm{prune}} \subset \mathcal{U}} \frac{1}{|\mathcal{D}_{\mathrm{cal}}|} \sum_{x \in \mathcal{D}_{\mathrm{cal}}} \Big( \mathcal{L}_{\mathrm{pred}}\big(f_{\mathrm{pruned}}(x)\big) - \mathcal{L}_{\mathrm{pred}}\big(f(x)\big) \Big);$$

where $f_{\mathrm{pruned}}$ denotes the model after removing the units in $\mathcal{U}_{\mathrm{prune}}$.

Model pruning can be performed either globally or layer-wise. Global pruning ranks all units across the model and removes the least important ones. Although theoretically effective, it often leads to uneven pruning, with some layers excessively pruned while others remain largely intact. Such imbalance can degrade model performance and complicate hardware deployment, making global pruning less common in practice, especially for large models with highly varying layer sensitivities. Consequently, in this work we adopt a layer-wise pruning strategy, commonly used in prior work (Sun et al., 2024; Ma et al., 2023; Ashkboos et al., 2024), and begin by examining how units within each individual layer contribute to the loss function.

### 3.2    LOSS-ALIGNED IMPORTANCE METRIC

Consider a pretrained model $f$ with $L$ layers, represented as the composition of layer-wise functions $f_1, f_2, \ldots, f_L$. Once the model parameters are fixed, each $f_l$ becomes a deterministic nonlinear mapping. Consequently, the entire network can be viewed as a fixed composition of such mappings.

Focusing on a particular layer $l$, the prediction $y_{\text{pred}}$ for an input $x$ can be expressed as:

$$y_{\text{pred}} = f(x) = \underbrace{(f_L \circ \cdots \circ f_{l+1})}_{\text{Subsequent sub-network } g_l} \circ \underbrace{(f_l \circ \cdots \circ f_1)(x)}_{\text{Output of layer } l \text{ denoted as } \boldsymbol{h}_l} = g_l(\boldsymbol{h}_l); \tag{1}$$

where $\boldsymbol{h}_l \in \mathbb{R}^d$ is the activation vector of layer $l$, and $g_l$ encapsulates the computation from layer $l+1$ to the output layer $L$. Let $\mathcal{U}_l$ denote the set of units in layer $l$, such as neurons or attention heads, and let $\boldsymbol{z}_u$ represent the contribution of unit $u \in \mathcal{U}_l$ to the layer's activation vector, then the layer activation can be expressed as $\boldsymbol{h}_l = \sum_{u \in \mathcal{U}_l} \boldsymbol{z}_u$.

For a single input $x$, the training loss of $f$ can be expressed as a function $\mathcal{L}$ of $\boldsymbol{h}_l$ as Eq. 2.

$$\mathcal{L}_{\text{pred}}(y_{\text{pred}}) = \mathcal{L}_{\text{pred}}(g_l(\boldsymbol{h}_l)) \equiv \mathcal{L}(\boldsymbol{h}_l) \tag{2}$$

Removing a subset of units $\mathcal{U}_l \subseteq \mathcal{U}_{\text{prune}}$ from layer $l$ is equivalent to perturbing its activation vector by $\Delta \boldsymbol{h}_l = -\sum_{u \in \mathcal{U}_l} \boldsymbol{z}_u$. Evaluating the impact of this perturbation on the loss of a single sample $x$, we expand $\mathcal{L}(\boldsymbol{h}_l)$ around the original activation using Taylor expansion:

$$\Delta \mathcal{L}(x) = \mathcal{L}(\boldsymbol{h}_l + \Delta \boldsymbol{h}_l) - \mathcal{L}(\boldsymbol{h}_l) \approx (\nabla_{\boldsymbol{h}_l} \mathcal{L})^\top \Delta \boldsymbol{h}_l + \frac{1}{2} \Delta \boldsymbol{h}_l^\top \nabla_{\boldsymbol{h}_l}^2 \mathcal{L} \, \Delta \boldsymbol{h}_l + O(\|\Delta \boldsymbol{h}_l\|^3); \tag{3}$$

where $\nabla_{\boldsymbol{h}_l} \mathcal{L}$ denotes the gradient of the loss with respect to $\boldsymbol{h}_l$. Although higher-order terms exist, they are intractable to compute and introduce complex cross-unit interactions that make the loss change non-additive. Hence, we restrict our analysis to the first-order approximation as Eq. 4.

$$\Delta \mathcal{L}(x) \approx (\nabla_{\boldsymbol{h}_l} \mathcal{L})^\top \Delta \boldsymbol{h}_l = \sum_{u \in \mathcal{U}_l} \left( -(\nabla_{\boldsymbol{h}_l} \mathcal{L})^\top \boldsymbol{z}_u \right) \tag{4}$$

For a single unit $u$, the induced loss change can be written as:

$$\Delta \mathcal{L}_u(x) \approx -(\nabla_{\boldsymbol{h}_l} \mathcal{L})^\top \boldsymbol{z}_u = -\frac{\partial \mathcal{L}}{\partial \boldsymbol{z}_u} \cdot \boldsymbol{z}_u; \tag{5}$$

where $\Delta \mathcal{L}_u(x)$ represents the loss change caused by removing a single unit $u$ and $\frac{\partial \mathcal{L}}{\partial \boldsymbol{z}_u}$ is the gradient of the loss with respect to the unit's output. Eq. 5 highlights that the importance of a unit depends jointly on its output magnitude and the sensitivity of the loss to that output.

By aggregating $\Delta \mathcal{L}_u(x)$ for unit $u$ over samples, we obtain the expected change in loss upon removing $u$ as Eq. 12, which defines the unit's importance metric for guiding pruning decisions.

$$\mu_u = \mathbb{E}_{x \sim \mathcal{D}_{\text{cal}}}[\Delta \mathcal{L}_u(x)] \tag{6}$$

### 3.3 Uncertainty-Aware Pruning

In practice, the loss change $\Delta \mathcal{L}_u(x)$ can exhibit highly skewed or heavy-tailed behavior, as detailed in Appendix D. In such cases, the empirical mean $\mu_u$ may underestimate the importance of the individual unit $u$. For instance, $\mu_u$ can be close to zero even when $u$ makes substantial contributions under rare but critical inputs. Moreover, the calibration dataset $\mathcal{D}_{\text{cal}}$ is typically limited in size, e.g., only hundreds of sequences, further limiting the confidence in the estimated importance.

Bernstein inequality (Audibert et al., 2007; Maurer & Pontil, 2009) implies that for any $\delta \in (0, 1)$, if $\Delta \mathcal{L}_u(x) : x \in \mathcal{D}_{\text{cal}}$ are independent and identically distributed and bounded in $[-1, 1]$, then the following holds:

$$\Pr\left( |\mu_u - m_u| \geq \sqrt{\frac{2 \ln(\frac{3}{\delta})}{|\mathcal{D}_{\text{cal}}|}} \cdot \sigma_u + \frac{6 \ln(\frac{3}{\delta})}{|\mathcal{D}_{\text{cal}}|} \right) \leq \delta; \tag{7}$$

where $m_u = \mathbb{E}_{x \sim \mathcal{D}}[\Delta \mathcal{L}_u(x)]$ is the true mean over the full underlying data distribution $\mathcal{D}$, as opposed to the empirical mean $\mu_u$ computed from the limited calibration dataset $\mathcal{D}\text{cal}$; $\sigma_u$ is the standard deviation of $\Delta \mathcal{L}_u(x)$ over the calibration dataset as $\sigma_u = \sqrt{\mathbb{E}_{x \sim \mathcal{D}_{\text{cal}}}\left[ \left( \Delta \mathcal{L}_u(x) - \mu_u \right)^2 \right]}$; and $\Pr(\cdot)$ denotes the probability of an event.

---

**Algorithm 1:** Loss-Aligned Structured Pruning

---

**Input:** Pretrained model $f$, calibration set $\mathcal{D}_{\text{cal}}$, pruning ratio $\gamma$, coefficient $\alpha$
**Output:** Pruned model $f_{\text{pruned}}$
**for** *each target layer $l$ in $f$* **do**
    Initialize score $s_u = 0$ for each unit $u \in \mathcal{U}_l$
    Initialize counter $n = 0$
    **for** *each sequence $x \in \mathcal{D}_{\text{cal}}$* **do**
        Forward $x$ through $f$ and compute activations at layer $l$
        Backward on $f$ to obtain the activation gradients of layer $l$
        Increment counter: $n \leftarrow n + 1$
        **for** *each unit $u \in \mathcal{U}_l$* **do**
            Calculate the sequence-level mean and standard deviation by Eq. 10 and Eq. 11
            Calculate sequence-level score $s_u(x) = \mu_u(x) + \alpha \cdot \sigma_u(x)$
            Update score using running average $s_u \leftarrow s_u + \frac{1}{n}(s_u(x) - s_u)$
    Rank units according to $s_u, u \in \mathcal{U}_l$ and prune the lowest $\lfloor \gamma \times |\mathcal{U}_l| \rfloor$ units

---

Rewriting the above inequality, with probability at least $1 - \delta$, the true mean $m_u$ is bounded by:

$$m_u \ \leq \ \mu_u + \sqrt{\frac{2\ln(\frac{3}{\delta})}{|\mathcal{D}_{\text{cal}}|}} \cdot \sigma_u + \frac{6\ln(\frac{3}{\delta})}{|\mathcal{D}_{\text{cal}}|} \tag{8}$$

Since the last term in this bound is constant across all units, it can be omitted when defining the importance metric for a unit. In practice, we adopt Bernstein-UCB, an optimistic strategy: if a unit's upper-bound importance is low, it is considered to be of low importance with high confidence.

UCB balances *exploitation* of high-reward options with *exploration* of uncertain ones by augmenting the mean estimate with an uncertainty-dependent bonus. A natural correspondence exists between this and the pruning problem: each unit can be treated as an option, where the expected loss reduction $\mu_u$ corresponds to its average reward, and the standard deviation of $\Delta\mathcal{L}_u(x)$ over inputs captures the uncertainty in that estimated reward. Consequently, the UCB enhanced importance score for neuron $u$ on sequence $x$ is then given by:

$$s_u \ = \ \mu_u + \alpha \cdot \sigma_u; \tag{9}$$

where $\alpha \geq 0$ is a tunable parameter that controls the emphasis on uncertainty. A larger $\alpha$ leads to a more conservative criterion, retaining neurons that may have low average contribution but high variability within the sequence, thereby ensuring high confidence when identifying low-importance units. The value of $\alpha$ can in principle be guided by Eq. 8; however, in practice it is often determined empirically, as done in many previous studies (Li et al., 2010; Liu et al., 2018; Li et al., 2024; 2025).

### 3.4 IMPLEMENTATION WORKFLOW

LASP proceeds sequentially over layers $1, 2, \ldots, L$ of $f$. In MLP layers, a unit corresponds to an individual neuron, whereas in attention layers, the granularity can vary, ranging from specific dimensions of queries, keys, or values to entire attention heads. In this work, for both efficiency and simplicity, we define a unit in attention layers as a complete attention head, meaning that pruning a unit entails removing the full set of associated query, key, and value projections. For each layer $l \in \{1, 2, \ldots, L\}$, the pruning procedure comprises two steps: computing the importance scores of units in the layer and removing those with the lowest importance according to the pruning ratio $\gamma$.

**Importance Score Computation**. In practice, the loss for each sequence $x \in \mathcal{D}_{\text{cal}}$ can be decomposed into token-level contributions, allowing a hierarchical formulation of unit importance. For a unit $u$, we compute its mean and standard deviation on a single sequence $x$ as:

$$\mu_u(x) = \frac{1}{|x|} \sum_{i=1}^{|x|} \left( - \frac{\partial \mathcal{L}(x_i)}{\partial \boldsymbol{z}_u(x_i)} \cdot \boldsymbol{z}_u(x_i) \right) \tag{10}$$

$$\sigma_u(x) = \sqrt{\frac{1}{|x|} \sum_{i=1}^{|x|} (-\frac{\partial \mathcal{L}(x_i)}{\partial \boldsymbol{z}_u(x_i)} \cdot \boldsymbol{z}_u(x_i) - \mu_u(x))^2} \tag{11}$$

The overall importance score $s_u$ of unit $u$ over the calibration dataset $\mathcal{D}_{\text{cal}}$ is then obtained by averaging sequence-level scores over all sequences:

$$s_u = \frac{1}{|\mathcal{D}_{\text{cal}}|} \sum_{x \in \mathcal{D}_{\text{cal}}} s_u(x) = \frac{1}{|\mathcal{D}_{\text{cal}}|} \sum_{x \in \mathcal{D}_{\text{cal}}} (\mu_u(x) + \alpha \cdot \sigma_u(x)) \tag{12}$$

This computation can be implemented efficiently using a moving average across the dataset, which provides stable estimates of both the mean and variance for each unit while reducing memory usage.

**Unit Pruning**. For both MLP and attention layers, units are ranked within each layer according to their importance scores. Given a pruning ratio $\gamma$, the units with the lowest scores are selected for removal. Pruning an MLP neuron corresponds to removing its associated row or column from the weight matrix, while pruning an attention unit corresponds to removing the complete attention head.

Algorithm 1 summarizes the pruning procedure. As we can see, the above two steps are applied independently to each layer, avoiding excessive degradation concentrated in a few critical layers.

## 4    EXPERIMENTS

In this section, we evaluate the effectiveness of LASP. We begin by describing the experimental settings, followed by performance evaluations, and conclude with ablation studies and further analyses.

### 4.1    EXPERIMENTAL SETTINGS

**Foundation LLMs**. To showcase the versatility of our method, we test it over three open-source LLM families that are widely used, including LLaMA model family (Touvron et al., 2023a), Vicuna-v1.5 model family (Zheng et al., 2023), and LLaMA2 model family (Touvron et al., 2023b).

**Calibration Dataset**. Following the settings of SliceGPT, we use 128 sequences of length 2048 sampled from the first shard of the WikiText2 training set (Merity et al., 2016) as our calibration dataset. Since WikiText2 is used for calibration, we evaluate the pruned models by conducting a perplexity analysis on its validation set.

**Evaluation Datasets**. To evaluate the model's performance in a task-agnostic setting, we apply the model pruned using the calibration dataset to zero-shot inference on several commonsense reasoning benchmarks, including BoolQ (Clark et al., 2019), PIQA (Bisk et al., 2020), HellaSwag (Zellers et al., 2019), WinoGrande (Sakaguchi et al., 2021), as well as ARC-Easy and ARC-Challenge (Clark et al., 2018). Evaluations are conducted using the lm-eval-harness library (Gao et al., 2021).

**Baseline Setup**. We compare our method with FLAP(An et al., 2024), LLM-Pruner(Ma et al., 2023) and two variants of SliceGPT (Ashkboos et al., 2024): the first is the original SliceGPT, which applies PCA-based structured pruning on the weight matrices, representing a typical structured pruning approach; the second is SliceGPT (w/ FT), a version of SliceGPT fine-tuned with LoRA (Hu et al., 2022) on 4,000 sequences from the Alpaca training set (Taori et al., 2023), each with a length of 1,024 tokens. To eliminate potential external factors—such as varying pruning ratios across layers with different sensitivities—all methods adopt a uniform pruning strategy, ensuring a fair and consistent comparison.

### 4.2    LANGUAGE MODELING

In this subsection, we present results on WikiText2 using the LLaMA-1, LLaMA-2, and Vicuna-v1.5 models. Table 1 reports the perplexity under pruning ratios of 20%, 25%, and 30%. Across all models, LASP consistently achieves lower perplexity than SliceGPT at the same pruning ratio. These results demonstrate that LASP preserves performance more effectively than SliceGPT under structured pruning across different model families.

### 4.3    ZERO-SHOT PERFORMANCE

Table 2 presents the results of LLaMA-2 7B and 13B models on zero-shot tasks under different pruning ratios, i.e., 20%, 25%, 30%. Across all tasks and pruning ratios, LASP consistently outperforms

Table 1: LLaMA-1, LLaMA-2, and Vicuna perplexity results on WikiText2 test set. The calibration set size and sequence length are 128 and 2048, respectively.

| Ratio | Method | LLaMA-1 | | | | LLaMA-2 | | Vicuna-v1.5 | |
|---|---|---|---|---|---|---|---|---|---|
| | | 7B | 13B | 30B | 65B | 7B | 13B | 7B | 13B |
| - | Dense | 5.47 | 4.88 | 4.10 | 3.53 | 5.47 | 4.88 | 6.78 | 5.95 |
| 20% | SliceGPT | 7.00 | 6.13 | 5.27 | 4.59 | 6.86 | 6.04 | 8.13 | 7.84 |
| | FLAP | 6.89 | 6.05 | 5.13 | 4.45 | 7.16 | 6.31 | 9.05 | 7.79 |
| | LASP | **6.18** | **5.38** | **4.54** | **3.98** | **6.16** | **5.38** | **6.86** | **5.91** |
| 25% | SliceGPT | 7.67 | 6.64 | 5.70 | 4.99 | 7.56 | 6.61 | 8.84 | 8.99 |
| | FLAP | 7.54 | 6.47 | 5.50 | 4.73 | 7.94 | 6.93 | 10.40 | 8.56 |
| | LASP | **6.75** | **5.72** | **4.83** | **4.23** | **6.73** | **5.72** | **7.46** | **6.95** |
| 30% | SliceGPT | 8.70 | 7.35 | 6.32 | 5.49 | 8.64 | 7.44 | 9.94 | 11.33 |
| | FLAP | 8.23 | 6.97 | 5.87 | 5.05 | 8.85 | 7.57 | 11.36 | 9.29 |
| | LASP | **7.28** | **6.18** | **5.14** | **4.54** | **7.29** | **6.16** | **7.98** | **6.86** |

Table 2: Evaluation results on multiple benchmarks.

| Model | Ratio | Method | ARC-c | ARC-e | BoolQ | HellaSwag | PIQA | Winogrande | Average |
|---|---|---|---|---|---|---|---|---|---|
| LLaMA-2 7B | - | Dense | 43.34 | 76.39 | 77.74 | 75.98 | 78.07 | 69.22 | 70.12 |
| | 20% | SliceGPT | 32.84 | 60.81 | 48.77 | 58.98 | 69.31 | 64.48 | 55.86 |
| | | FLAP | 30.97 | 63.80 | 63.08 | 64.77 | 73.72 | 63.77 | 60.02 |
| | | LLM-Pruner | 23.80 | 57.28 | 63.09 | 44.06 | 68.82 | 53.59 | 51.77 |
| | | LASP | **40.70** | **73.32** | **70.15** | **71.36** | **76.93** | **65.35** | **66.30** |
| | 25% | SliceGPT | 32.25 | 61.23 | 51.52 | 54.31 | 66.21 | 62.90 | 54.73 |
| | | FLAP | 30.54 | 60.43 | 50.36 | 60.44 | 71.70 | 61.56 | 55.84 |
| | | LLM-Pruner | 17.74 | 43.47 | 60.03 | 33.05 | 62.35 | 50.51 | 44.52 |
| | | LASP | **39.08** | **70.45** | **67.40** | **66.62** | **75.30** | **61.09** | **63.32** |
| | 30% | SliceGPT | 29.01 | 55.93 | 38.59 | 49.10 | 63.32 | 62.66 | 49.76 |
| | | FLAP | 28.83 | 59.51 | 44.74 | 56.55 | 69.64 | 62.03 | 53.55 |
| | | LLM-Pruner | 17.74 | 37.58 | 56.08 | 30.57 | 59.14 | 48.14 | 41.54 |
| | | LASP | **36.18** | **69.78** | **64.34** | **63.90** | **73.45** | **59.35** | **61.16** |
| LLaMA-2 13B | - | Dense | 48.29 | 79.42 | 80.58 | 79.37 | 79.16 | 72.14 | 73.16 |
| | 20% | SliceGPT | 38.65 | 71.25 | 44.95 | 62.79 | 70.78 | 67.56 | 59.33 |
| | | FLAP | 37.88 | 70.07 | 67.52 | 67.78 | 74.15 | 66.77 | 64.03 |
| | | LLM-Pruner | 34.89 | 69.86 | 67.95 | 63.45 | 75.02 | 59.66 | 61.80 |
| | | LASP | **41.72** | **73.74** | **70.34** | **75.95** | **77.58** | **68.67** | **67.93** |
| | 25% | SliceGPT | 35.83 | 65.69 | 40.88 | 57.39 | 68.00 | **68.19** | 56.00 |
| | | FLAP | 35.92 | 67.17 | 65.44 | 64.69 | 72.90 | 66.29 | 62.07 |
| | | LLM-Pruner | 29.52 | 66.49 | 63.27 | 52.45 | 71.05 | 56.43 | 56.54 |
| | | LASP | **41.72** | **74.62** | **75.96** | **74.22** | **76.82** | 66.85 | **68.37** |
| | 30% | SliceGPT | 32.50 | 59.42 | 38.74 | 52.16 | 64.47 | **65.58** | 52.15 |
| | | FLAP | 34.55 | 62.20 | 65.20 | 61.88 | 71.38 | **65.58** | 60.13 |
| | | LLM-Pruner | 23.46 | 57.61 | 62.11 | 41.86 | 65.88 | 53.75 | 50.78 |
| | | LASP | **38.74** | **72.69** | **72.26** | **70.52** | **74.76** | 63.61 | **65.43** |

SliceGPT. While the performance of both methods improves with model size, LASP maintains a clear advantage. Notably, the 7B model pruned by 25% and the 13B model pruned by 30% still retain around 90% of the original dense model's average accuracy. Moreover, for both model sizes, LASP at 30% pruning ratio surpasses SliceGPT at 20%, highlighting its ability to preserve model quality even under aggressive pruning.

Furthermore, Fig. 2 presents a direct comparison between our method and the fine-tuned SliceGPT. For the LLaMA-2 7B model, our method consistently outperforms SliceGPT across all pruning ratios. For the LLaMA-2 13B model, it performs slightly worse than the fine-tuned SliceGPT at

20% and 30% pruning, but surpasses it at 25% pruning. Overall, across both model scales, our method is comparable to or slightly better than the fine-tuned SliceGPT.

## 4.4 ABLATION STUDY

In this subsection, we conduct an ablation study on the hyperparameter $\alpha$ of the uncertainty term. As shown in Fig. 3, the perplexity curves for both LLaMA-2 7B and 13B models exhibit a U-shape, confirming that an optimal balance between *exploitation* and *exploration* is essential.

Setting the hyperparameter $\alpha$ to a small value, e.g., $\alpha = 0.01$, unit selection is driven almost entirely by the average first-order loss change. As $\alpha$ increases, the standard-deviation term is progressively incorporated, encoding confidence in each neuron's estimated loss contribution and improving the performance. According to Fig. 3, this improvement peaks when the confidence term is weighted enough to guide the neuron selection. However, increasing $\alpha$ beyond this point begins to degrade performance due to scale mismatch. As shown in Appendix D, the standard deviation is much larger in magnitude than the average loss. Consequently, when $\alpha$ becomes sufficiently large, e.g., $\alpha = 0.5$, the importance score becomes dominated by the variance, ignoring the mean contribution, which harms fluency and increases perplexity.

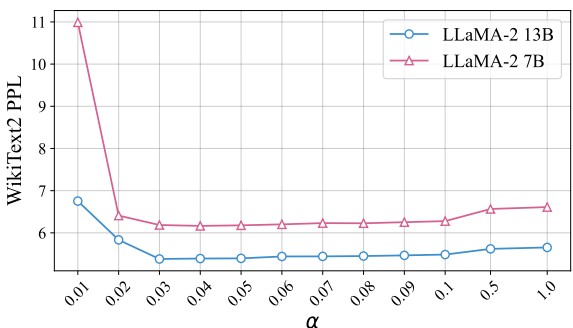

Figure 3: Ablation study of hyperparameter $\alpha$, illustrating how the performance of LASP varies with different values of $\alpha$ for LLaMA-2 7B and 13B models.

Overall, these observations underscore that choosing $\alpha$ carefully is essential for balancing mean contribution and confidence. Ideally, $\alpha$ should be set so that the mean and variance terms operate on comparable scales, ensuring that neither dominates the importance score.

## 4.5 MORE ANALYSIS

**Data sensitivity**. We examine how calibration data size and sequence length affect pruning performance, as shown in Fig. 4. When the sequence length is fixed at 2048, increasing the calibration set size consistently lowers perplexity. This improvement arises from more accurate estimates of both the mean first-order loss change and its standard deviation; with more samples, these statistics better approximate their true underlying distribution, leading to more reliable importance scores. As the sample size grows, the perplexity curve begins to plateau, indicating that these estimates have become sufficiently stable. When fixing the number of samples to 128 and varying the sequence

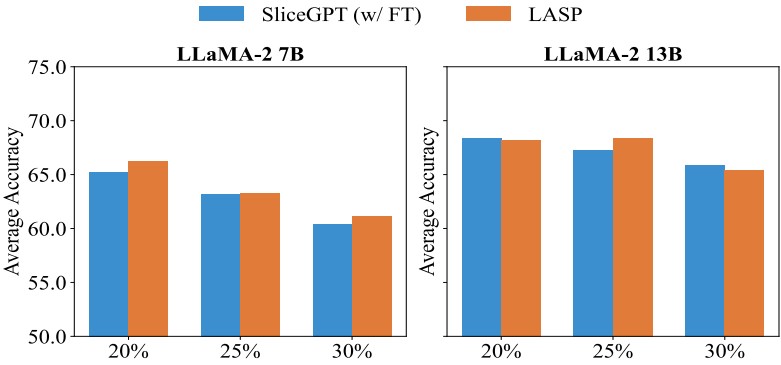

Figure 2: Average accuracy of SliceGPT (w/ FT) and LASP, which is not fine-tuned, across pruning ratios of 20%, 25%, and 30% on LLaMA-2 7B and 13B models over seven benchmark tasks.

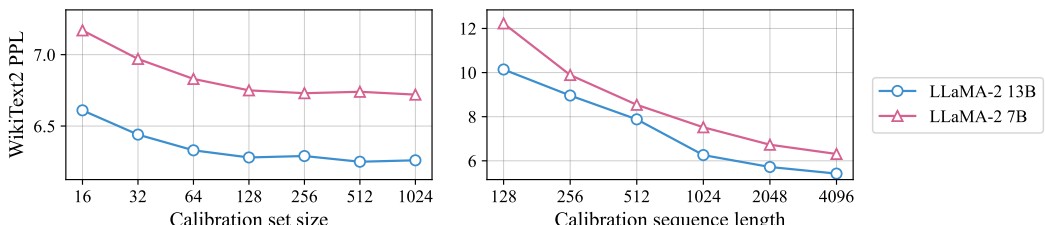

Figure 4: Effect of the calibration set size and sequence length on perplexity of WikiText2.

length, perplexity also decreases. This reduction is driven not only by the inherent advantage of modeling longer contexts, but also by the fact that longer sequences contain more token-level observations per sample. These additional observations make each sample's importance estimate more precise, yielding better pruning decisions and lower perplexity. Overall, both trends demonstrate that richer calibration data—either through more samples or longer sequences—produces more stable importance estimation and thus improves pruning performance.

**Pruning dynamics**. To better understand how our method works on each layer, we analyze the pruning dynamics of LLaMA2-7B under different pruning ratios. Fig. 5 illustrates how different pruning ratios affect the model's average loss across layers. A clear trend emerges: the impact of pruning varies substantially with network depth. In the early layers, roughly layers 1 to 5, pruning consistently lowers the loss, revealing a high level of parameter redundancy. A pruning ratio of 0.20 already leads to a noticeable reduction in loss, suggesting that removing redundant neurons functions as a form of regularization and improves generalization. The middle layers, around layers 8 to 18, show the opposite behavior. Redundancy is minimal, and these layers play a central role in maintaining predictive accuracy. Even moderate pruning here produces an increase in loss, indicating that this region forms a critical bottleneck for model performance. In the later layers, approximately layers 18 to 32, a moderate degree of redundancy reappears. Pruning can be applied with limited adverse effects, although excessive removal still degrades performance. Taken together, the redundancy pattern across the network is highly structured: strongest in the early layers, weakest in the middle, and moderate toward the end.

Another key observation concerns the redundancy in the model's internal representations. Pruning the first few layers results in similar loss values across different pruning ratios, reflecting substantial redundancy in these layers. Extending pruning to additional layers reveals a more nuanced pattern. For example, applying a 0.20 pruning ratio across up to ten layers achieves lower loss than pruning only five or six layers at higher ratios, such as 0.25 or 0.30. This suggests that the model has an optimal capacity for its internal representations, thus excessive pruning reduces the amount of meaningful information passed to subsequent layers, limiting the gains from further pruning and potentially harming per-

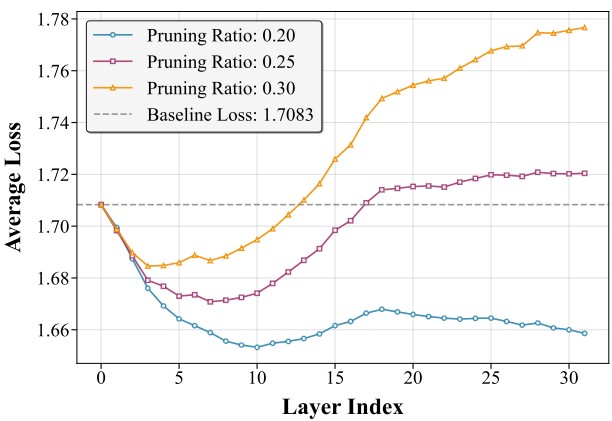

Figure 5: Pruning dynamics of LLaMA-7B model. Curves with different colors represent different pruning ratios.

formance. These findings underscore the critical balance between structural redundancy and information propagation, highlighting that effective pruning must consider not only the number of neurons removed but also the integrity of information flow throughout the network.

Moreover, Fig. 5 further reveals an intriguing phenomenon: when the pruning ratio is set to 0.20, the loss of the pruned model becomes even lower than that of the pre-trained LLM. This counter-

intuitive improvement provides additional evidence that our approach is inherently loss-aligned. Rather than removing capacity, the pruning procedure selectively eliminates redundant or uninformative neurons, effectively denoising internal representations and enhancing the efficiency of information flow. Consequently, the model not only maintains its expressiveness but can also achieve marginally improved optimization behavior, illustrating that, when properly applied, pruning serves not only as a compression technique but also as a beneficial inductive guide for the model.

## 5 CONCLUSION

In this work, we have proposed LASP, a loss-aligned structured pruning method for LLMs that directly evaluates the contribution of model units to overall performance. By integrating activation magnitudes with gradients and leveraging the UCB strategy, LASP effectively identifies and removes low-importance units while mitigating uncertainty from limited calibration data. Our implementation further reduces storage overhead through running statistics. Extensive experiments across multiple LLMs and benchmarks demonstrate that LASP consistently outperforms state-of-the-art pruning methods, achieving a favorable trade-off between efficiency and model performance. These results highlight the potential of LASP for enabling the practical deployment of LLMs without significant loss in predictive capability. Beyond empirical effectiveness, our approach provides a new perspective by modeling the loss directly at the unit-output level. We hope that this formulation will inspire further theoretical insights into LLM pruning.

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

## A  LLM USAGE

In this work, large language models are mainly used as research assistants to support literature exploration and text refinement. Specifically, we leverage LLMs to (i) improve the clarity and conciseness of our writing, and (ii) assist in locating relevant conference papers on pruning methods, including both unstructured and structured approaches, and locating key survey articles concerning the broader topics of model compression, pruning, and quantization.

## B  IMPLEMENTATION OF USING SELF-SUPERVISED LOSS

### B.1  THEORY ANALYSIS

We aim to evaluate the change in loss $\Delta\mathcal{L}$ caused by pruning a unit across the entire calibration dataset $\mathcal{D}_{\text{cal}}$. Formally, this is defined as the average change over all sequences in $\mathcal{D}_{\text{cal}}$:

$$\Delta\mathcal{L}_u = \frac{1}{|\mathcal{D}_{\text{cal}}|} \sum_{x \in \mathcal{D}_{\text{cal}}} \Delta\mathcal{L}_u(x). \tag{13}$$

To understand this average, we first analyze the loss on a single sequence $x$. In maximum likelihood training, the loss on a sequence $\mathbf{x} = (x_1, x_2, \ldots, x_{|x|})$ is defined as the negative log-likelihood, which can be expressed as the average of token-level losses:

$$\mathcal{L}(x) = -\frac{1}{|x|} \sum_{i=1}^{|x|} \log p(x_i \mid x_{<i}), \tag{14}$$

where $|x|$ is the length of the sequence.

Accordingly, the loss change on $x$ caused by pruning a unit is

$$\Delta\mathcal{L}_u(x) \approx \frac{1}{|x|} \sum_{i=1}^{|x|} \left( -\frac{\partial \log p(x_i \mid x_{<i})}{\partial \mathbf{z}_u(x_i)} \cdot \mathbf{z}_u(x_i) \right), \tag{15}$$

According to our algorthm, we calculate $\mu_u(x)$ and $\sigma_u(x)$ to get the importance score $s_u(x)$:

$$\mu_u(x) \approx \frac{1}{|x|} \sum_{i=1}^{|x|} \left( -\frac{\partial \log p(x_i \mid x_{<i})}{\partial \mathbf{z}_u(x_i)} \cdot \mathbf{z}_u(x_i) \right), \tag{16}$$

$$\sigma_u(x) \approx \sqrt{\frac{1}{|x|} \sum_{i=1}^{|x|} \left( \left( -\frac{\partial \log p(x_i \mid x_{<i})}{\partial \mathbf{z}_u(x_i)} \cdot \mathbf{z}_u(x_i) \right) - \mu_u(x) \right)^2}. \tag{17}$$

By applying our algorithm to all sequences in the calibration dataset $\mathcal{D}_{\text{cal}}$, we obtain the overall importance score $S_u$:

$$S_u = \frac{1}{|\mathcal{D}_{\text{cal}}|} \sum_{x \in \mathcal{D}_{\text{cal}}} s_u(x). \tag{18}$$

For each layer, after obtaining the overall scores $\{S_u\}$, we sort all units in ascending order according to $S_u$, and then prune the lowest-ranked units according to the target pruning ratio $r$.

### B.2  IMPLEMENTATION DETAILS

In practice, for the mean term in the significance score, we directly take the empirical average over the calibration dataset. For the $\sigma$ term, since it is simultaneously affected by both the sample length (which is fixed across all pruning samples) and the hyperparameter $\alpha$, we merge these two factors into a single adjustable coefficient $\alpha$ for simplicity. This treatment keeps the implementation convenient while retaining the flexibility of controlling the variance penalty. For our approach, the best coefficient $\alpha$ under different pruning ratios can be seen in Table 3.

Table 3: coefficient $\alpha$ settings for different models under various pruning ratios.

| Model | 20% | 25% | 30% |
|-------|-----|-----|-----|
| llama2-7b | 0.03 | 0.03 | 0.03 |
| llama2-13b | 0.03 | 0.03 | 0.04 |
| llama-7b | 0.03 | 0.05 | 0.03 |
| llama-13b | 0.03 | 0.03 | 0.03 |
| llama-30b | 0.07 | 0.07 | 0.07 |
| llama-65b | 0.12 | 0.10 | 0.12 |
| vicuna-v1.5-7b | 0.03 | 0.03 | 0.03 |
| vicuna-v1.5-13b | 0.03 | 0.0117 | 0.0175 |

Moreover, for finetuning the SliceGPT-pruned LLaMA-2 model, we employ LoRA for efficient adaptation. Specifically, we set the training batch size to 3, with LoRA-$\alpha = 10$, rank $r = 32$, and a dropout rate of 0.05. LoRA modules are injected into both attention heads and MLP layers, enabling parameter-efficient fine-tuning while maintaining the performance of the pruned model.

## C    EFFICIENCY ANALYSIS.

Table 4 reports the efficiency statistics of the pruned LLaMA2-13B models under different pruning ratios (PR), including the number of multiply–accumulate operations (MACs), runtime peak memory consumption, inference latency, and model load memory. The reported MACs correspond to the prefill stage, while memory and latency are measured during the generation of 1024 tokens. As the pruning ratio increases, both the computational requirements and memory footprint decrease consistently. For example, at a pruning ratio of 20%, MACs are reduced from 822.64G to 660.25G, runtime memory drops from 25.7 GiB to 20.7 GiB, and inference latency improves from 32.75s to 27.73s. At 25% pruning, further efficiency gains are observed, with memory usage reduced to 19.5 GiB and latency to 26.10s. When the pruning ratio reaches 30%, latency remains stable at 25.89s, while both MACs and memory continue to decline, with runtime memory reduced to 18.2 GiB. These results show that structured pruning substantially reduces computation and memory costs while delivering more efficient inference without introducing instability in runtime behavior. (All results are measured on the WikiText2 test set using a single NVIDIA RTX 4090 48G GPU.)

Table 4: Efficiency statistics of the pruned models under different pruning ratios (PR).

| PR (%) | MACs (G) | Runtime Memory (MiB) | Latency (s) | Model Load Memory (MiB) |
|--------|----------|----------------------|-------------|-------------------------|
| 0% | 822.644 | 25748.890 | 32.753 | 24826.792 |
| 20% | 660.254 | 20726.856 | 27.727 | 19994.854 |
| 25% | 619.618 | 19470.430 | 26.100 | 18783.917 |
| 30% | 579.020 | 18215.437 | 25.891 | 17574.151 |

## D    FIRST-ORDER LOSS VALUE VISUALIZATION

As Fig. 6 and Fig. 7 show, the first-order loss approximation exhibits a heavy-tailed distribution: for more than 50% of the tokens within a sequence, the value is close to zero, while in a few cases a neuron plays a critical role and yields a large loss change.

Therefore, relying solely on the empirical mean of $\Delta\mathcal{L}_u(\mathbf{x})$ observed on a calibration set $\mathcal{D}_{\text{cal}}$ to estimate a unit's importance is therefore risky. On the one hand, the limited size of $\mathcal{D}_{\text{cal}}$ introduces uncertainty into the estimation; on the other hand, the heavy-tailed nature of $\Delta\mathcal{L}_u$ means that the mean is often dominated by near-zero values, overlooking the substantial influence a neuron may exert on a small subset of critical tokens. To address this, we introduce UCB, which explicitly accounts for rare but significant variations and improves confidence in assessing a unit's influence, thereby enabling more accurate selection.

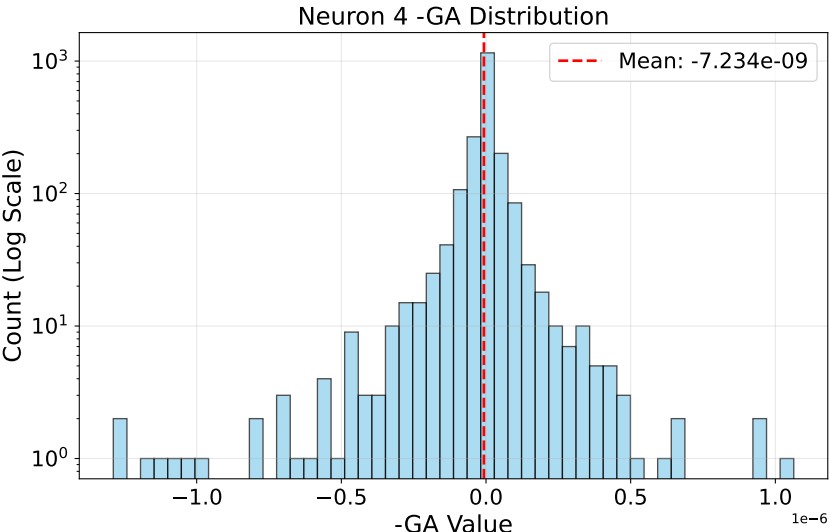

Figure 6: Distribution of $-\mathrm{GA}$ values for the 4-th neuron in the first MLP layer. Most values concentrate near zero, while a few significant deviations highlight the necessity of incorporating uncertainty into the pruning criterion.

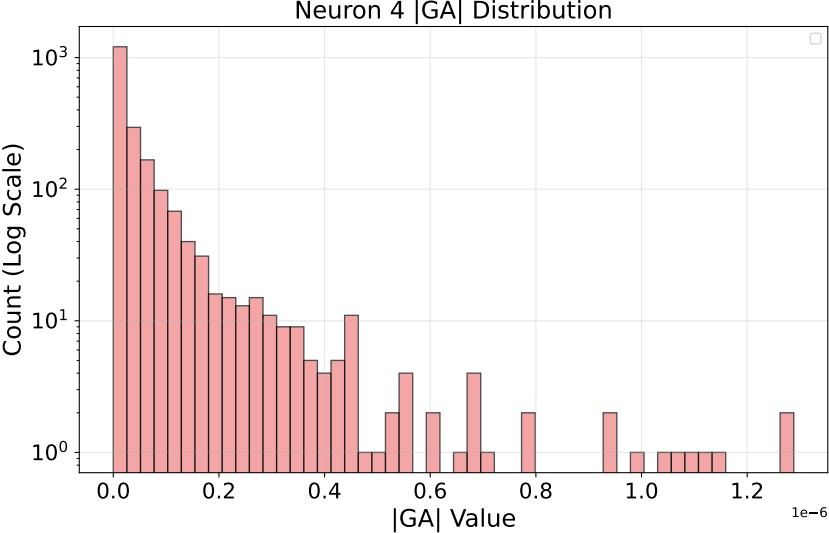

Figure 7: Distribution of $|GA|$ values for the 4-th neuron in the first MLP layer. The distribution in the figure clearly shows a long-tail distribution.

