# OpenReview forum: "Loss-Aligned Structured Pruning for Large Language Models"
_ICLR.cc/2026/Conference — Submitted to ICLR 2026_

### Official Review · Reviewer_sGME · 2025-10-23

**Soundness:** 2
**Presentation:** 2
**Contribution:** 2
**Rating:** 2
**Confidence:** 3

**Summary:**

The paper proposes a loss-aligned structured pruning approach for large language models (LLMs). The key idea is to align pruning decisions (i.e., which structures—in this case groups of weights/rows/columns—are removed) with their measured impact on the loss (or proxy thereof) rather than purely on magnitude or heuristic importance.

**Strengths:**

The paper proposes a loss-aligned pruning metric that directly correlates with the model’s final task loss, which is more task-relevant than using only activations or gradients. The inclusion of uncertainty modeling via an Upper Confidence Bound (UCB) is relatively novel in the pruning literature and provides a principled way to handle estimation uncertainty. The method is empirically validated across Llama and some datasets, demonstrating a good balance between pruning efficiency and performance retention.

**Weaknesses:**

1. Testing should be conducted on newer models, such as the Qwen3 series. Moreover, whether the method remains effective on reasoning models is also an open question.
2. The evaluation datasets are insufficient. The current datasets are too simple, and the pruned parameters might not cover the activation regions of these models. More diverse and challenging datasets should be included.
3. The proposed method feels too incremental and lacks sufficient novelty or contribution.
4. In addition, the paper is missing important citations that should be discussed, such as SparseGPT [1], LoraPrune [2], Compresso [3], and LLM-Pruner [4].
References:
[1] SparseGPT: Massive Language Models Can Be Accurately Pruned in One-Shot
[2] LoraPrune: Pruning Meets Low-Rank Parameter-Efficient Fine-Tuning
[3] Compresso: Structured Pruning with Collaborative Prompting Learns Compact Large Language Models
[4] LLM-Pruner: On the Structural Pruning of Large Language Models

**Questions:**

Same as weakness.

---

> ### Author Response · Authors · 2025-11-23
> **Response to Reviewer sGME(1/2)**
>
> __Q1:__ Testing should be conducted on newer models, such as the Qwen3 series. Moreover, whether the method remains effective on reasoning models is also an open question.
>
> __A1:__ Thank you for your suggestions.
>
> Our evaluation process follows the **standard setting** of prominent baselines such as SliceGPT [1], LLM-Pruner [2], and FLAP [3], which primarily use the LLaMA series for the experiments. Accordingly, we adopt this setting to ensure both comparability with prior work and consistency in benchmarking. Moreover, to demonstrate the generality of our method, we additionally conduct experiments on Vicuna models, where our approach likewise achieves performance superior to the baselines.
>
> Following your suggestion, we conducted **additional experiments** on the Qwen2.5-7B model. Using the same pruning setup described in the paper (for LLM-Pruner, we use param-second mode and skip the first and last 4 layers). During the pruning process we observed that our method remains **loss-aligned**. On the WikiText-2 calibration set, the original loss was 1.93, while the post-pruning losses was 2.04 at 20% sparsity.
>
> The result table below also indicate that our pruning  method introduces only minimal performance degradation, i.e., it preserves the loss behavior and most of the ability of the original model, even on Qwen2.5-7B. This further shows that the effectiveness of our approach extends beyond the LLaMA architecture family.
>
> **Table:Evaluation results of pruned Qwen2.5-7B model**
> | Ratio | Method | Perplexity      | ARC-c | ARC-e | BoolQ | HellaSwag | PIQA | Winogrande |
> |-------|-------|-------------|--------|--------|--------|------------|-------|-------------|
> | Dense | - | 6.85 | 47.78 | 80.42 | 84.71 | 78.93 | 78.78 | 70.93 |
> | 20% | LASP | 10.48 | 37.80 | 68.01 | 62.08 | 58.94 | 70.84 | 61.48 |
> |  | LLM-Pruner | 231.67 | 20.22 | 32.70 | 40.92 | 31.11 | 57.13 | 50.67 |
> | 25% | LASP | 15.18 | 33.53 | 61.11 | 62.20 | 51.49 | 67.19 | 58.40 |
> |  | LLM-Pruner | 522.07 | 20.22 | 32.70 | 40.92 | 31.11 | 57.13 | 50.67 |
>
> Overall, these experiments verify the generality of our model in two aspects:
> -  LLaMA models of different scales and variants exhibit distinct activation dynamics and pruning sensitivities, yet our method remains effective across them.
> -  Additional experiments on Qwen2.5 model, which utilize GQA mechanism, demonstrate that our method remains effective across different language model architectures.
>
> Regarding the application to reasoning models, we acknowledge that this remains a largely unexplored area in current structured pruning research. Existing methods rely on calibration data to identify important units, meaning the selected units are inherently data-dependent. While obtaining high-quality reasoning data is challenging, once such data becomes available, our method can be readily applied to reasoning models under the same framework.
>
> [1] Ashkboos S, Croci M L, Nascimento M G, et al. Slicegpt: Compress large language models by deleting rows and columns. ICLR 2024.
>
> [2] Ma X, Fang G, Wang X. LLM-Pruner: On the structural pruning of large language models. Neurips 2023.
>
> [3] An Y, Zhao X, Yu T, et al. Fluctuation-based adaptive structured pruning for large language models. AAAI 2024.
>
> __Q2:__ The evaluation datasets are insufficient. The current datasets are too simple, and the pruned parameters might not cover the activation regions of these models. More diverse and challenging datasets should be included.
>
> __A2:__ Thanks for your suggestions.
>
> **Our evaluation follows the standard pipeline adopted by well-known methods such as LLM-Pruner, SliceGPT, and FLAP, which is adopted by the most of pruning articles**. We selected a diverse set of widely used reasoning benchmarks, e.g., ARC, BoolQ, HellaSwag, PIQA, Winogrande, to ensure fair and direct comparison with existing pruning studies. Based on this, we believe **our benchmark selection is appropriate for the purpose of evaluating structured pruning methods**.
>
> We would also like to note that prior pruning works generally focus on verifying whether a method can prune effectively given a calibration dataset. The construction or selection of the datasets is largely **an orthogonal direction**, and improving the dataset side is outside the primary scope of structured pruning research.
>
> That said, we sincerely appreciate the reviewer’s perspective and agree that extending pruning studies to more complex datasets is a valuable and under-explored direction for future work.

---

> ### Author Response · Authors · 2025-11-23
> **Response to Reviewer sGME(2/2)**
>
> __Q3:__ The proposed method feels too incremental and lacks sufficient novelty or contribution.
>
> __A3:__ Thanks for your question.
>
> We would like to once again clarify the design rationale behind our method and highlight how it differs from existing baselines.
>
> 1. In contrast to our approach, SliceGPT focuses solely on minimizing the reconstruction loss at each individual layer, without directly considering the performance loss of the pruned model. This layer-wise focus does not explicitly guarantee the overall model performance, **as the outputs of individual layers may be amplified or diminished by subsequent layers**. By not directly linking pruning to the model’s final performance, SliceGPT’s method may fail to preserve the pruned model’s effectiveness, whereas our approach explicitly optimizes for post-pruning performance and demonstrates superior results.
> 2. In contrast to gradient-based methods like LLM‑Pruner, their approach approximates the loss at the **parameter level** and then aggregates these scores to estimate the importance of larger structures. It relies on constructing a dependency graph to identify coupled neurons that must be pruned together. While this ensures safe structured pruning, it introduces **implementation overhead** and **reduces flexibility**, as the coupled neurons can no longer be independently pruned based on their individual contributions. Experimental results also show that pruning dependent neurons together is less effective than evaluating and pruning each unit individually.
> 3. In contrast, our method evaluates the importance of **each unit** based on its contribution to minimizing the model’s loss and examines its distribution across samples. By accounting for both the mean effect and associated uncertainty, the UCB score enables us to confidently identify units that may have rare but high-impact contributions.
>
> __Q4:__ In addition, the paper is missing important citations that should be discussed, such as SparseGPT [1], LoraPrune [2], Compresso [3], and LLM-Pruner [4]. References: [1] SparseGPT: Massive Language Models Can Be Accurately Pruned in One-Shot [2] LoraPrune: Pruning Meets Low-Rank Parameter-Efficient Fine-Tuning [3] Compresso: Structured Pruning with Collaborative Prompting Learns Compact Large Language Models [4] LLM-Pruner: On the Structural Pruning of Large Language Models.
>
> __A4:__ Thanks for the suggestion.
>
> **In our first submission, We have already cited and discussed LLM-Pruner and SparseGPT (both in Related Work).**
>
> Discussion of SparseGPT in our first submission : "The SparseGPT method (Frantar & Alistarh, 2023) offers a solution by developing a more efficient weight update procedure for LLMs, which uses synchronized second-order Hessian updates
> to circumvent this computational bottleneck."
>
> Discussion of LLM-Pruner in our first submission: "LLM-Pruner (Ma et al., 2023), on the other hand, evaluates the importance of
> each parameter using gradient and second-order information, aggregates these scores at the channel or neuron level, and prunes accordingly to enable efficient deployment of large models."
>
> **Follow your suggestion, we have carefully compared these works (Compresso, LoRA-Prune, etc.), and the corresponding discussion has been added to the revised version.**

---

> > ### Comment · Reviewer_sGME · 2025-11-28
> > **Response to authors**
> >
> > Thank you for the authors’ response. I will first raise my rating to 4. However, as the capabilities of the reasoning model (similar to Qwen3) have not been verified and I am not an expert in this area, I will make my final recommendation based on the comments of the other reviewers.

---

> ### Author Response · Authors · 2025-11-28
> **Response to Reviewer sGME**
>
> We sincerely thank the reviewer for the thoughtful feedback and for raising the score.
>
> To address your remaining concern regarding the reasoning capabilities of the model, we have conducted additional experiments on Deepseek-R1-distilled-7B model, which is known for its strong reasoning abilities. We changed the calibration set to the first 1024 samples in ServiceNow-AI/R1-Distill-SFT dataset and compared our method with LLM-Pruner. After the pruning process we found that the loss drops from 0.47 to 0.40, indicating that our method continually aligns with the model's loss objective. We have also tested the pruned model's performance on diverse math benchmarks, including gsm8k, mmlu_college_mathematics, mmlu_high_school_mathematics, and mmlu_high_school_statistics. The results are summarized in the table below.
>
> **Table:Evaluation results of pruned Deepseek-R1-distilled-7B model**
> | Ratio | Method | gsm8k | mmlu_college_mathematics | mmlu_high_school_mathematics | mmlu_high_school_statistics |
> |-------|-------|--------|--------|--------|------------|
> |  | - | 19.94 | 41.00 | 44.70 | 61.11  |
> | 20% | LASP | 16.75 | 36.00 | 41.48 | 45.37  |
> |  | LLM-Pruner | 3.00 | 28.00 | 21.85 | 24.07 |
>
> Our updated results on math benchmarks consistently show that our method continues to outperform existing baselines, demonstrating its effectiveness on reasoning models as well.
>
> We hope that this new evidence helps alleviate your concern. Thank you again for your time and constructive comments.

---

### Official Review · Reviewer_nRoF · 2025-10-26

**Soundness:** 2
**Presentation:** 3
**Contribution:** 3
**Rating:** 2
**Confidence:** 5

**Summary:**

This paper presents LASP (Loss-Aligned Structured Pruning), a method for efficient compression of Large Language Models (LLMs) through structured pruning. Instead of reconstructing weight matrices or estimating second-order Hessians, LASP introduces a first-order, loss-aligned importance metric that evaluates neuron or head significance using activation–gradient interactions directly tied to model loss. To address data uncertainty from small calibration sets, the method integrates an Upper Confidence Bound (UCB) strategy and employs moving-average running statistics to minimize storage overhead. Experiments on LLaMA, LLaMA-2, and Vicuna models show that LASP achieves superior performance retention—up to 93.5% at 25% pruning—compared with state-of-the-art baselines such as SliceGPT.

**Strengths:**

1. No fine-tuning required: LASP achieves strong performance preservation without any fine-tuning, making it highly practical for large-scale deployment where retraining costs are prohibitive.
2. More accurate pruning criterion: By aligning pruning decisions with loss gradients rather than proxy metrics like weight magnitude or reconstruction error, LASP offers a theoretically grounded and empirically superior estimation of unit importance.
3. Engineering and algorithmic efficiency: The paper introduces concrete optimizations—layer-wise backward computation, running statistics, and small calibration sets—to make loss-based pruning computationally feasible on single GPUs, effectively addressing the global loss alignment cost.

**Weaknesses:**

1. Missing comparison with FLAP (AAAI 2024): Since the paper focuses on pruning without fine-tuning (w/o FT), it is crucial to include comparisons with FLAP, which also targets the same setting, to better contextualize LASP’s contributions.
2. Limited model diversity: The evaluation is restricted to the LLaMA and Vicuna series. Experiments on more diverse architectures such as DeepSeek or Qwen would strengthen the generality claims.
3. Lack of pruning-time analysis: Given that the motivation is to avoid fine-tuning costs, it is equally important to report the actual pruning runtime and computational resources, to validate that the method indeed offers an overall efficiency gain.
4. Missing w/ FT evaluation: Although the paper emphasizes the w/o FT setting, fine-tuning typically incurs a moderate cost compared to pretraining. Including w/ FT experiments would reveal the upper bound of model performance and provide a fuller understanding of LASP’s potential.

[1] Fluctuation-based Adaptive Structured Pruning for Large Language Models. AAAI 2024

**Questions:**

See weaknesses

---

> ### Author Response · Authors · 2025-11-23
> **Response to Reviewer nRoF(1/3)**
>
> __Q1:__ Missing comparison with FLAP (AAAI 2024): Since the paper focuses on pruning without fine-tuning (w/o FT), it is crucial to include comparisons with FLAP, which also targets the same setting, to better contextualize LASP’s contributions.
>
> __A1:__ Thanks for the sugguestion.
>
> We choose SliceGPT as the baseline because it is a representative and powerful pruning method. Although FLAP and SliceGPT were proposed around the same time, SliceGPT was formally accepted slightly later, making it the more up-to-date approach. Therefore, we select SliceGPT as the primary baseline for comparison.
>
> Following your suggestion, we conducted additional experiments of FLAP. The results, summarized below, show that our method consistently outperforms FLAP across all models at 20%, 25%, and 30% sparsity. Specifically, averaged across multiple benchmarks, LASP-pruned LLaMA-2 7B and 13B models achieve **6%–8%** and **4%–6%** higher performance, respectively, compared to the second-best method under the same sparsity levels. This consistent improvement further demonstrates the robustness and generality of our approach.
>
> We have incorporated these results into the revised submission.
>
> __Table: LLaMA-1, LLaMA-2, and Vicuna perplexity results on WikiText2 test set__
>
> | Ratio | Method   | LLaMA-1 7B | 13B  | 30B  | 65B  | LLaMA-2 7B | 13B  | Vicuna 7B | 13B  |
> |-------|----------|------------|------|------|------|------------|------|-----------|------|
> | -     | Dense    | 5.47       | 4.88 | 4.10 | 3.53 | 5.47       | 4.88 | 6.78      | 5.95 |
> | 20%   | FLAP     | 6.89       | 6.05 | 5.13 | 4.45 | 7.16       | 6.31 | 9.05      | 7.79 |
> |       | LASP     | **6.18**   | **5.38** | **4.54** | **3.98** | **6.16** | **5.38** | **6.86** | **5.91** |
> | 25%   | FLAP     | 7.54       | 6.47 | 5.50 | 4.73 | 7.94       | 6.93 | 10.40     | 8.56 |
> |       | LASP     | **6.75**   | **5.72** | **4.83** | **4.23** | **6.73** | **5.72** | **7.46** | **6.95** |
> | 30%   | FLAP     | 8.23       | 6.97 | 5.87 | 5.05 | 8.85       | 7.57 | 11.36     | 9.29 |
> |       | LASP     | **7.28**   | **6.18** | **5.14** | **4.54** | **7.29** | **6.16** | **7.98** | **6.86** |
>
> __Table: Evaluation results on multiple benchmarks.__
>
> | __Model__ | __Ratio__ | __Method__      | __ARC-c__ | __ARC-e__ | __BoolQ__ | __HellaSwag__ | __PIQA__ | __Winogrande__ | __Average__ |
> |-------|-------|-------------|--------|--------|--------|------------|-------|-------------|----------|
> | LLaMA-2 7B | - | Dense | 43.34 | 76.39 | 77.74 | 75.98 | 78.07 | 69.22 | 70.12 |
> | | 20% | FLAP | 30.97 | 63.80 | 63.08 | 64.77 | 73.72 | 63.77 | 60.02 |
> | |     | LASP       | **40.70** | **73.32** | **70.15** | **71.36** | **76.93** | **65.35** | **66.30** |
> | | 25% | FLAP | 30.54 | 60.43 | 50.36 | 60.44 | 71.70 | **61.56** | 55.84 |
> | |     | LASP       | **39.08** | **70.45** | **67.40** | **66.62** | **75.30** | 61.09 | **63.32** |
> | | 30% | FLAP | 28.83 | 59.51 | 44.74 | 56.55 | 69.64 | **62.03** | 53.55 |
> | |     | LASP       | **36.18** | **69.78** | **64.34** | **63.90** | **73.45** | 59.35 | **61.16** |
> | LLaMA-2 13B | - | Dense | 48.29 | 79.42 | 80.58 | 79.37 | 79.16 | 72.14 | 73.16 |
> | | 20% | FLAP | 37.88 | 70.07 | 67.52 | 67.78 | 74.15 | 66.77 | 64.03 |
> | |     | LASP       | **41.72** | **73.74** | **70.34** | **75.95** | **77.58** | **68.67** | **67.93** |
> | | 25% | FLAP | 35.92 | 67.17 | 65.44 | 64.69 | 72.90 | 66.29 | 62.07 |
> | |     | LASP       | **41.72** | **74.62** | **75.96** | **74.22** | **76.82** | **66.85** | **68.37** |
> | | 30% | FLAP | 34.55 | 62.20 | 65.20 | 61.88 | 71.38 | **65.58** | 60.13 |
> | |     | LASP       | **38.74** | **72.69** | **72.26** | **70.52** | **74.76** | 63.61 | **65.43** |

---

> ### Author Response · Authors · 2025-11-23
> **Response to Reviewer nRoF(2/3)**
>
> __Q2:__ Limited model diversity: The evaluation is restricted to the LLaMA and Vicuna series. Experiments on more diverse architectures such as DeepSeek or Qwen would strengthen the generality claims.
>
> __A2:__ Thank you for your suggestions.
>
> Our evaluation process follows the **standard setting** of prominent baselines such as SliceGPT [1], LLM-Pruner [2], and FLAP [3], which primarily use the LLaMA series for the experiments. Accordingly, we adopt this setting to ensure both comparability with prior work and consistency in benchmarking. Moreover, to demonstrate the generality of our method, we additionally conduct experiments on Vicuna models, where our approach likewise achieves performance superior to the baselines.
>
> Following your suggestion, we conducted **additional experiments** on the Qwen2.5-7B model. Using the same pruning setup described in the paper (for LLM-Pruner, we use param-second mode and skip the first and last 4 layers). During the pruning process we observed that our method remains **loss-aligned**. On the WikiText-2 calibration set, the original loss was 1.93, while the post-pruning losses was 2.04 at 20% sparsity.
>
> The result table below also indicate that our pruning  method introduces only minimal performance degradation, i.e., it preserves the loss behavior and most of the ability of the original model, even on Qwen2.5-7B. This further shows that the effectiveness of our approach extends beyond the LLaMA architecture family.
>
> **Table:Evaluation results of pruned Qwen2.5-7B model**
> | Ratio | Method | Perplexity      | ARC-c | ARC-e | BoolQ | HellaSwag | PIQA | Winogrande |
> |-------|-------|-------------|--------|--------|--------|------------|-------|-------------|
> | Dense | - | 6.85 | 47.78 | 80.42 | 84.71 | 78.93 | 78.78 | 70.93 |
> | 20% | LASP | 10.48 | 37.80 | 68.01 | 62.08 | 58.94 | 70.84 | 61.48 |
> |  | LLM-Pruner | 231.67 | 20.22 | 32.70 | 40.92 | 31.11 | 57.13 | 50.67 |
> | 25% | LASP | 15.18 | 33.53 | 61.11 | 62.20 | 51.49 | 67.19 | 58.40 |
> |  | LLM-Pruner | 522.07 | 20.22 | 32.70 | 40.92 | 31.11 | 57.13 | 50.67 |
>
> Overall, these experiments verify the generality of our model in two aspects:
> -  LLaMA models of different scales and variants exhibit distinct activation dynamics and pruning sensitivities, yet our method remains effective across them.
> -  Additional experiments on Qwen2.5 model, which utilize GQA mechanism, demonstrate that our method remains effective across different language model architectures.
>
>
> [1] Ashkboos S, Croci M L, Nascimento M G, et al. Slicegpt: Compress large language models by deleting rows and columns. ICLR, 2024.
>
> [2] Ma X, Fang G, Wang X. LLM-Pruner: On the structural pruning of large language models. Neurips, 2023.
>
> [3] An Y, Zhao X, Yu T, et al. Fluctuation-based adaptive structured pruning for large language models. AAAI, 2024.
>
> __Q3:__ Lack of pruning-time analysis: Given that the motivation is to avoid fine-tuning costs, it is equally important to report the actual pruning runtime and computational resources, to validate that the method indeed offers an overall efficiency gain.
>
> __A3:__ thanks for your suggestion.
>
> According to your suggestion, we have collected the pruning costs of our method and summarize it in the table below. Besides, if you are interested in efficiency gains after pruning, our first submission have provided it in Appendix C.
>
> __Table: Resource cost analysis of pruning__
> |  __Model__        | __Method__       | __Peak GPU Memory (GB)__ | __Pruning Time (min)__ |
> |--------------|--------------|------------------------|----------------------|
> | LLaMA-2 7B   | LASP   |       21.31            |      17.96           |
> | LLaMA-2 13B  | LASP   |       37.46            |      47.15           |
>
> As shown in the results table, our method **remains straightforward to run on consumer-grade GPUs**. For instance, the LLaMA-2 7B model can be fully pruned on a 24 GB RTX 4090 GPU, demonstrating the accessibility and efficiency of our approach. Moreover, the pruning time is of limited practical concern: (1) what ultimately matters is the model’s efficiency and performance after pruning, and (2) the extra overhead is small, fully acceptable, and well within a reasonable range for practical deployment.

---

> ### Author Response · Authors · 2025-11-23
> **Response to Reviewer nRoF(3/3)**
>
> __Q4:__ Missing w/ FT evaluation: Although the paper emphasizes the w/o FT setting, fine-tuning typically incurs a moderate cost compared to pretraining. Including w/ FT experiments would reveal the upper bound of model performance and provide a fuller understanding of LASP’s potential.
>
> __A4:__ Thanks for your suggestion.
>
> According to our results, our method preserves most of the model’s performance, indicating that in practice it is often unnecessary to fine-tune the pruned model.
>
> Following your suggestion, we fine-tuned both the LASP-pruned model and the SliceGPT-pruned model using LoRA, adopting the same hyperparameter settings as in the SliceGPT paper and leveraging the first 8,000 samples from the Alpaca training set. As shown in the results table below, **our method already achieves performance comparable to the LoRA-fine-tuned SliceGPT model even without additional fine-tuning**, and further improves after applying LoRA.
>
> This demonstrates that, in terms of both cost-effectiveness and final performance, our approach offers clear advantages over the baseline.
>
>
> __Table: Performance of pruned LLaMA-2 7B model after LoRA finetuning on multiple benchmarks__
>
> | __Model__ | __Ratio__ | __Method__      | __ARC-c__ | __ARC-e__ | __BoolQ__ | __HellaSwag__ | __PIQA__ | __Winogrande__ | __Average__ |
> |-------|-------|-------------|--------|--------|--------|------------|-------|-------------|----------|
> | LLaMA-2 7B | - | Dense | 43.34 | 76.39 | 77.74 | 75.98 | 78.07 | 69.22 | 70.12 |
> | | 20% | LASP w/o ft | 40.70 | 73.32 | 70.15 | 71.36 | 76.93 | 64.48 | 66.30 |
> | |     | LASP w ft       | 42.83 | 74.32 | 74.58 | 71.47| 77.96 | 66.21 | 67.90 |
> | |     | SliceGPT w ft       | 40.18 | 69.53 | 70.12 | 66.73 | 73.12 | 64.09 | 63.96 |
> | | 25% | LASP w/o ft | 39.08 | 70.45 | 67.40 | 66.62 | 75.30 | 61.09 | 63.32 |
> | |     | LASP w ft       | 42.74 | 73.95 | 74.34 | 71.42 | 78.13 | 66.06 | 67.77 |
> | |     | SliceGPT w ft       | 38.31 | 66.79 | 67.95 | 62.61 | 71.11 | 64.17 | 61.82 |
> | | 30% | LASP w/o ft | 36.18 | 69.78 | 64.34 | 63.90 | 73.45 | 59.35 | 61.16 |
> | |     | LASP w ft       | 39.07 | 70.87 | 71.03 | 66.19 | 75.46 | 62.35 | 64.16 |
> | |     | SliceGPT w ft       | 36.43 | 64.81 | 66.97 | 59.21 | 68.93 | 62.27 | 59.77 |

---

> ### Author Response · Authors · 2025-11-28
> **We would be grateful if you could take a look at the response**
>
> To further show LASP's generality, we have extended our experiment on Deepseek-R1-distilled-7B model, which is known for its strong reasoning abilities. We changed the calibration set to the first 1024 samples in ServiceNow-AI/R1-Distill-SFT dataset and compared our method with LLM-Pruner. The results are summarized in the table below.
>
> **Table:Evaluation results of pruned Deepseek-R1-distilled-7B model**
> | Ratio | Method | gsm8k | mmlu_college_mathematics | mmlu_high_school_mathematics | mmlu_high_school_statistics |
> |-------|-------|--------|--------|--------|------------|
> |  | - | 19.94 | 41.00 | 44.70 | 61.11  |
> | 20% | LASP | 16.75 | 36.00 | 41.48 | 45.37  |
> |  | LLM-Pruner | 3.00 | 28.00 | 21.85 | 24.07 |
>
> Our updated results on math benchmarks consistently show that our method continues to outperform existing baselines, demonstrating its effectiveness on reasoning models as well.
>
> We sincerely appreciate your valuable time devoted to reviewing our manuscript. We would like to gently remind you of the approaching deadline for the discussion phase. We have diligently addressed the issues you raised in your feedback, providing detailed explanations. For instance, we:
> - Provided evaluation results on Qwen2.5-7B to demonstrate the generality of our method across different architectures.
> - Included comparisons with FLAP (AAAI 2024) in both perplexity and downstream tasks to contextualize our contributions.
> - Added a comprehensive runtime and computational cost analysis to clarify the efficiency.
>  Would you kindly take a moment to look at it? We are very enthusiastic about engaging in more in-depth discussions with you.
>
> For ease of reference, we also list the specific changes below:
> - Related work: FLAP (An et al., 2024) takes a pioneering step by using fluctuation-based strategy that dynamically adapts
> pruning decisions, offering a promising direction for structured pruning under high sparsity.
> - Experiments (language modeling section): Beyond SliceGPT, we also compare with FLAP, which demonstrates improved perplexity performance across most model sizes. Nonetheless, our proposed LASP consistently achieves lower perplexity than FLAP across all model variants and pruning ratios, confirming the effectiveness of directly aligning unit importance with the model’s loss objective.

---

### Official Review · Reviewer_6ggD · 2025-10-28

**Soundness:** 2
**Presentation:** 3
**Contribution:** 2
**Rating:** 4
**Confidence:** 3

**Summary:**

This paper proposes LASP (Loss-Aligned Structured Pruning), a post-training structured pruning framework for large language models (LLMs). LASP evaluates the importance of structural units (neurons or attention heads) by combining their activation magnitudes with gradients of the loss function, thereby aligning pruning decisions with the model’s training objective. To mitigate uncertainty from limited calibration data, LASP introduces an Upper Confidence Bound (UCB)–style regularization that accounts for variance in importance estimates. Empirical results on several open-source models (LLaMA, LLaMA2, Vicuna) show consistent performance improvements over SliceGPT across perplexity and zero-shot reasoning benchmarks. The method is computationally lightweight and hardware-friendly, requiring no retraining.

**Strengths:**

+ Clear and intuitive formulation. The paper presents a well-motivated first-order approximation of loss change and integrates uncertainty modeling via UCB in a mathematically coherent way. The approach is conceptually simple and easy to implement.

+ Strong experimental coverage. Results are reported for multiple LLM families and scales, with pruning ratios up to 30%. The evaluation includes both perplexity (WikiText-2) and zero-shot reasoning tasks, showing consistent trends.

+ Practicality. LASP is post-training and structured, making it appealing for real-world deployment where retraining is infeasible. Efficiency improvements in memory and latency are quantified.

**Weaknesses:**

+ Limited novelty relative to prior structured pruning methods. While the “loss-aligned” formulation is conceptually elegant, the underlying idea—using first-order gradient × activation statistics to rank units—is closely related to long-standing importance-based pruning and Taylor expansion–based saliency measures. The use of UCB for uncertainty modeling is a mild novelty but not deeply justified or theoretically analyzed. Overall, LASP reads as an incremental extension of SliceGPT or LLM-Pruner rather than a substantive conceptual advance.

+ Inadequate comparative evaluation. The experimental section primarily contrasts LASP with SliceGPT (with and without LoRA fine-tuning). However, other strong baselines—SparseGPT, LLM-Pruner, or WoodFisher—are discussed but not empirically compared. This omission weakens the claim of “state-of-the-art” performance, especially since SparseGPT has demonstrated superior accuracy-efficiency trade-offs under similar data constraints.

+ Weak theoretical justification for UCB integration. The introduction of the UCB term appears heuristic. There is no formal connection drawn between the multi-armed bandit setting (where UCB is derived) and the pruning uncertainty scenario here. Moreover, the hyperparameter α controlling the uncertainty trade-off is tuned empirically without principled guidance or sensitivity analysis beyond a basic ablation (Figure 3).

+ Lack of interpretability or mechanistic analysis. The paper provides no insight into which neurons or attention heads are pruned or retained, nor any qualitative discussion of redundancy patterns beyond superficial loss plots. This limits understanding of why LASP works and whether its pruning behavior is generalizable across architectures.

+ Presentation and clarity issues. The exposition is generally clear but occasionally repetitive (e.g., re-deriving standard first-order approximations). Figures could better highlight empirical differences between LASP and baselines. Some notation (e.g., the definition of ∆Lu) is inconsistent across sections, and the ablation analysis is relatively shallow.

**Questions:**

+ Could you formalize the connection between UCB and pruning uncertainty beyond analogy? Is there a principled reason to expect the UCB bonus to approximate the true risk of pruning a unit?

+ Why were LLM-Pruner and SparseGPT omitted from empirical comparison, given their relevance and accessibility?

+ Can LASP be extended to finer-grained structures (e.g., Q/K/V sub-dimensions or per-channel pruning in MLPs), and would the loss-alignment principle still hold?

+ How robust is α across architectures and tasks? Could you propose a heuristic or scaling rule to set α without exhaustive tuning?

---

> ### Author Response · Authors · 2025-11-23
> **Response to Reviewer 6ggD(1/4)**
>
> __Q1:__ Limited novelty relative to prior structured pruning methods. While the “loss-aligned” formulation is conceptually elegant, the underlying idea—using first-order gradient × activation statistics to rank units—is closely related to long-standing importance-based pruning and Taylor expansion–based saliency measures. The use of UCB for uncertainty modeling is a mild novelty but not deeply justified or theoretically analyzed. Overall, LASP reads as an incremental extension of SliceGPT or LLM-Pruner rather than a substantive conceptual advance.
>
> __A1:__ Thank you for recognizing the conceptual elegance of our loss-aligned idea.
>
> We would like to once again clarify the design rationale behind our method and highlight how it differs from existing baselines.
>
> 1. In contrast to our approach, SliceGPT focuses solely on minimizing the reconstruction loss at each individual layer, without directly considering the performance loss of the pruned model. This layer-wise focus does not explicitly guarantee the overall model performance, **as the outputs of individual layers may be amplified or diminished by subsequent layers**. By not directly linking pruning to the model’s final performance, SliceGPT’s method may fail to preserve the pruned model’s effectiveness, whereas our approach explicitly optimizes for post-pruning performance and demonstrates superior results.
> 2. In contrast to gradient-based methods like LLM‑Pruner, their approach approximates the loss at the **parameter level** and then aggregates these scores to estimate the importance of larger structures. It relies on constructing a dependency graph to identify coupled neurons that must be pruned together. While this ensures safe structured pruning, it introduces **implementation overhead** and **reduces flexibility**, as the coupled neurons can no longer be independently pruned based on their individual contributions. Experimental results also show that pruning dependent neurons together is less effective than evaluating and pruning each unit individually.
> 3. In contrast, our method evaluates the importance of **each unit** based on its contribution to minimizing the model’s loss and examines its distribution across samples. By accounting for both the mean effect and associated uncertainty, the UCB score enables us to confidently identify units that may have rare but high-impact contributions.

---

> > ### Author Response · Authors · 2025-11-28
> > **We would be grateful if you could take a look at the response**
> >
> > We sincerely appreciate your valuable time devoted to reviewing our manuscript. We would like to gently remind you of the approaching deadline for the discussion phase. We have diligently addressed the issues you raised in your feedback, providing detailed explanations. For instance, we
> > - Compared LASP with FLAP and LLM-Pruner to demonstrate superior performance across various model sizes and pruning ratios.
> > - Formalized the connection between our method and the traditional UCB theory.
> > - Demonstrated our method's effectiveness when extended to fine-grained structure.
> > - Provided a theoretical bound and discussed about how to select the hyperparameter $\alpha$.
> >
> > Would you kindly take a moment to look at it? We are very enthusiastic about engaging in more in-depth discussions with you.

---

> ### Author Response · Authors · 2025-11-23
> **Response to Reviewer 6ggD(2/4)**
>
> __Q2:__ Inadequate comparative evaluation. The experimental section primarily contrasts LASP with SliceGPT. However, other strong baselines—SparseGPT, LLM-Pruner, or WoodFisher—are discussed but not empirically compared. This omission weakens the claim of “state-of-the-art” performance, especially since SparseGPT has demonstrated superior accuracy-efficiency trade-offs under similar data constraints.
>
> __A2:__ Thank you for your suggestion.
>
> The goal of our study is to develop a **structured pruning method**, whose advantages over unstructured pruning approaches have been thoroughly discussed in the paper.
>
> **SparseGPT is an unstructured pruning method**, which is orthogonal to our research focus on structured pruning. **WoodFisher, proposed in 2020, is also an unstructured pruning method**. We select SliceGPT as our primary baseline, as it is a recent, representative, and strong structured pruning approach.
>
> Following your suggestion, we have additionally included LLM-Pruner [1] and FLAP [2] as baselines in our extended experiments. As shown in the following table, our method consistently achieves the best performance among all compared approaches.
>
>
> __Table: Evaluation results on multiple benchmarks.__
>
> | Model | Ratio | Method      | ARC-c | ARC-e | BoolQ | HellaSwag | PIQA | Winogrande | Average |
> |-------|-------|-------------|--------|--------|--------|------------|-------|-------------|----------|
> | LLaMA-2 7B | - | Dense | 43.34 | 76.39 | 77.74 | 75.98 | 78.07 | 69.22 | 70.12 |
> | | 20% | FLAP | 30.97 | 63.80 | 63.08 | 64.77 | 73.72 | 63.77 | 60.02 |
> | |     | LLM-Pruner       | 23.80 | 57.28 | 63.09 | 44.06 | 68.82 | 53.59 | 51.77 |
> | |     | LASP       | **40.70** | **73.32** | **70.15** | **71.36** | **76.93** | **65.35** | **66.30** |
> | | 25% | FLAP | 30.54 | 60.43 | 50.36 | 60.44 | 71.70 | **61.56** | 55.84 |
> | |     | LLM-Pruner       | 17.74 | 43.47 | 60.03 | 33.05 | 62.35 | 50.51 | 44.52 |
> | |     | LASP       | **39.08** | **70.45** | **67.40** | **66.62** | **75.30** | 61.09 | **63.32** |
> | | 30% | FLAP | 28.83 | 59.51 | 44.74 | 56.55 | 69.64 | **62.03** | 53.55 |
> | |     | LLM-Pruner       | 17.74 | 37.58 | 56.08 | 30.57 | 59.14 | 48.14 | 41.54 |
> | |     | LASP       | **36.18** | **69.78** | **64.34** | **63.90** | **73.45** | 59.35 | **61.16** |
> | LLaMA-2 13B | - | Dense | 48.29 | 79.42 | 80.58 | 79.37 | 79.16 | 72.14 | 73.16 |
> | | 20% | FLAP | 37.88 | 70.07 | 67.52 | 67.78 | 74.15 | 66.77 | 64.03 |
> | |     | LLM-Pruner       | 34.89 | 69.86 | 67.95 | 63.45 | 75.02 | 59.66 | 61.80 |
> | |     | LASP       | **41.72** | **73.74** | **70.34** | **75.95** | **77.58** | **68.67** | **67.93** |
> | | 25% | FLAP | 35.92 | 67.17 | 65.44 | 64.69 | 72.90 | 66.29 | 62.07 |
> | |     | LLM-Pruner       | 29.52 | 66.49 | 63.27 | 52.45 | 71.05 | 56.43 | 56.54 |
> | |     | LASP       | **41.72** | **74.62** | **75.96** | **74.22** | **76.82** | **66.85** | **68.37** |
> | | 30% | FLAP | 34.55 | 62.20 | 65.20 | 61.88 | 71.38 | **65.58** | 60.13 |
> | |     | LLM-Pruner       | 23.46 | 57.61 | 62.11 | 41.86 | 65.88 | 53.75 | 50.78 |
> | |     | LASP       | **38.74** | **72.69** | **72.26** | **70.52** | **74.76** | 63.61 | **65.43** |
>
> These new results have been incorporated into the revised version of our paper.
>
> [1] Ma X, Fang G, Wang X. LLM-Pruner: On the structural pruning of large language models. Neurips, 2023.
>
> [2] An Y, Zhao X, Yu T, et al. Fluctuation-based adaptive structured pruning for large language models. AAAI, 2024.

---

> ### Author Response · Authors · 2025-11-23
> **Response to Reviewer 6ggD(3/4)**
>
> __Q3:__ Weak theoretical justification for UCB integration. The introduction of the UCB term appears heuristic. There is no formal connection drawn between the multi-armed bandit setting and the pruning uncertainty scenario here. Moreover, the hyperparameter α controlling the uncertainty trade-off is tuned empirically without principled guidance or sensitivity analysis beyond a basic ablation.
>
> __A3:__ Thanks for your question.
>
> **1. About the theoretical justification:**
> The traditional stochastic multi-armed bandit setting assumes that each arm corresponds to an unknown reward distribution, which can only be accessed through repeated sampling.
>
> In our work, we formulate neuron pruning as a stochastic multi-armed bandit problem: each unit is treated as an arm, and its importance score for a given input is modeled as a sample drawn from an unknown distribution. We adopt the **Bernstein-UCB formulation** [3] in this setting.
>
> Bernstein inequality implies that for any $\delta \in (0,1)$, the following holds:
>
> $$
> \Pr \left( \left| \mu_u - m_u \right| \ge \sqrt{\frac{2 \ln(\frac{3}{\delta})}{|\mathcal{D}\mathrm{cal}|}} \cdot \sigma_u + \frac{6 \ln(\frac{3}{\delta})}{|\mathcal{D}\mathrm{cal}|} \right) \le \delta
> $$
>
> where $m_u$ is the true mean over the full underlying data distribution, as opposed to the empirical mean $\mu_u$ computed from the limited calibration dataset; $\sigma_u$ is the standard deviation over the calibration dataset. The UCB strategy in our method LASP is derived from it. **We have added the detailed derivation in subsection 3.3 of the revised submission**.
>
> **2. About the setting of hyperparameter $\alpha$:**
> According to the **Bernstein inequality**, we derived the setting of $\alpha = \sqrt{\frac{2 \ln\\left(\frac{3}{\delta}\right)}{|\mathcal{D}_\mathrm{cal}|}}$ in subsection 3.3 of the revised paper. In practice, the value of $\alpha$ is usaully tuned emprically [4][5].
>
> Our ablation study demonstrates that the performance of LASP is robust within a broad range of $\alpha$. Across LLaMA-1 series, LLaMA-2-7B/13B, Vicuna, and Qwen3-8B, sweeping $\alpha$ in a reasonably wide range like 0.01 to 0.1 leads to only minor performance variation, this suggests that **$\alpha$ does not require extensive fine-grained tuning for each architecture.**
>
> [3] Jean-Yves Audibert, et al. Tuning bandit algorithms in stochastic environments. ALT, 2007.
>
> [4] Lihong Li, et al. A contextual-bandit approach to personalized news article recommendation. WWW, 2010.
>
> [5] Yexin Li, et al. A contextual combinatorial bandit approach to negotiation. ICML, 2024.
>
>
> __Q4:__ Lack of interpretability or mechanistic analysis. The paper provides no insight into which neurons or attention heads are pruned or retained, nor any qualitative discussion of redundancy patterns beyond superficial loss plots. This limits understanding of why LASP works and whether its pruning behavior is generalizable across architectures.
>
> __A4:__ Thanks for your questions.
>
> As described in subsection 3.4 of our paper, our method performs layer-wise pruning, where each layer is pruned **using the same pruning ratio, independent of redundancy patterns, which enables better generalization across different architectures**. Given that LLMs consist of millions of neurons and attention heads, fine-grained analysis of exactly which neurons or heads are pruned is impractical and provides limited practical insight. Insights into which neurons or attention heads are pruned or retained can be effectively understood through the **importance metric** proposed in our paper. For a more detailed theoretical explanation, please refer to __A3__.
>
> Consistent with representative structured pruning works such as LLM-Pruner, SliceGPT, and FLAP, which also place less emphasis on explicitly modeling redundancy, our method similarly focuses on analyzing meaningful metrics to guide pruning decisions.
>
> We acknowledge that detailed redundancy analysis is **an orthogonal research direction**. It can complement many pruning methods, including ours, and can be integrated with LASP without conceptual conflict. However, it is not the primary focus of this work. We plan to further explore this promising direction in future research.

---

> ### Author Response · Authors · 2025-11-23
> **Response to Reviewer 6ggD(4/4)**
>
> __Q5:__ Presentation and clarity issues. The exposition is generally clear but occasionally repetitive, e.g., re-deriving standard first-order approximations. Figures could better highlight empirical differences between LASP and baselines. Some notation is inconsistent across sections, and the ablation analysis is relatively shallow.
>
> __A5:__ thank for the feedback, **We have corrected these issues in the revised version.**
>
>
> __Q6:__ Could you formalize the connection between UCB and pruning uncertainty beyond analogy? Is there a principled reason to expect the UCB bonus to approximate the true risk of pruning a unit?
>
> __A6:__ thanks for the question.
>
> For the detailed mechanism explanation, please refer to the answer __A3__.
>
> In the revised version, we have also provided a theoretical analysis and derivation of the UCB term to further clarify the theoretical foundation of our method.
>
>
> __Q7:__ Why were LLM-Pruner and SparseGPT omitted from empirical comparison, given their relevance and accessibility?
>
> __A7:__ Thanks for the question.
>
> **SparseGPT employs a diagonal approximation of the Hessian to estimate the importance of individual weights**, and **is a typical unstructured pruning method.** which is orthogonal to our research.
>
> **For comparison with LLM‑Pruner, detailed comparison is provided in __A2__. Please refer to it.**
>
>
> __Q8:__ Can LASP be extended to finer-grained structures, e.g., Q/K/V sub-dimensions or per-channel pruning in MLPs, and would the loss-alignment principle still hold?
>
> __A8:__ Thanks for the question.
>
> **Our method is a general and flexible method, which can be applied to any structure within the network.** So surely it can be extended to QKV sub-dimensions and per-channel pruning in MLPs. However, we do not recommend pruning sub-dimensions of Q/K/V, as doing so can cause inconsistencies with the RoPE positional embedding mechanism.
>
> To further validate the generality of our approach, we extended our experiments to the Qwen2.5-7B model and set the attention's pruning granularity to the query in GQA. **Under the uniform pruning setting, we observed that on the WikiText-2 calibration set, the loss-alignment principle still hold**, for example: Pruning the first ten layers reduces the calibration loss from 1.93 to 1.92,  while the post-pruning loss at 20% sparsity is 2.04.
>
> These results confirm that our method remains effective even when the pruning granularity becomes smaller, demonstrating its robustness and adaptability.
>
> __Q9:__ How robust is $\alpha$ across architectures and tasks? Could you propose a heuristic or scaling rule to set $\alpha$ without exhaustive tuning?
>
> __A9:__ Thanks for the question.
>
> According to our expriment, we find that LASP is robust to the choice of $\alpha$ within a certain range. Across LLaMA-1 series, LLaMA-2-7B/13B, Vicuna, and Qwen2.5-7B, sweeping $\alpha$ in a reasonably wide range like 0.01 to 0.1 leads to only minor performance variation, this suggests that **$\alpha$ does not require extensive fine-grained tuning for each architecture.**
>
> In __A3__, we also provide a theoretically motivated setting for $\alpha$. However, for practical use, we recommend the empirical tuning strategy described above, as it is simple, effective, and robust across different architectures.

---

### Official Review · Reviewer_uyTN · 2025-11-05

**Soundness:** 3
**Presentation:** 3
**Contribution:** 2
**Rating:** 4
**Confidence:** 3

**Summary:**

This paper proposes LASP (Loss-Aligned Structured Pruning), a post-training structured pruning method for compressing large language models efficiently. LASP departs from reconstruction- or Hessian-based approaches by introducing a first-order, loss-aligned importance metric, which measures neuron or attention head importance through activation–gradient correlations directly tied to loss behavior. To mitigate calibration data uncertainty, LASP integrates a UCB-based exploration strategy and moving-average statistics to stabilize importance estimation while keeping storage overhead low. Experiments on LLaMA, LLaMA-2, and Vicuna models demonstrate that LASP maintains up to 93.5% performance at 25% pruning, outperforming SliceGPT and other pruning baselines under the no fine-tuning (w/o FT) setting.

**Strengths:**

(1) The proposed activation–gradient-based importance metric provides a principled and effective way to measure pruning sensitivity, improving upon traditional weight magnitude or reconstruction heuristics.
(2) The framework introduces practical engineering optimizations—layer-wise backward computation and running statistics—that make loss-based pruning computationally feasible even on limited hardware.
(3) LASP preserves strong performance in the w/o FT setting, significantly reducing deployment overhead for large models.

**Weaknesses:**

(1) Evaluation focuses on LLaMA and Vicuna only; broader validation on architectures such as DeepSeek or Qwen would better support claims of generality.
(2) Missing comparison with FLAP (AAAI 2024), as FLAP also operates under the w/o FT scenario, omitting this baseline limits contextual understanding of LASP’s contribution.
(3) The paper does not provide detailed runtime or computational cost analysis, leaving unclear whether pruning itself offers net efficiency gains.
(4 )While the paper emphasizes the w/o FT regime, including w/ FT results could highlight the potential performance ceiling and help assess trade-offs more comprehensively.

**Questions:**

What are the actual pruning-time and resource costs, and how do they compare with existing efficient pruning baselines?
Would including fine-tuned (w/ FT) results clarify LASP’s headroom and performance scalability?

---

> ### Author Response · Authors · 2025-11-23
> **Response to Reviewer uyTN(1/3)**
>
> __Q1:__ Evaluation focuses on LLaMA and Vicuna only; broader validation on architectures such as DeepSeek or Qwen would better support claims of generality.
>
> __A1:__ Thank you for your suggestions.
>
> Our evaluation process follows the **standard setting** of prominent baselines such as SliceGPT [1], LLM-Pruner [2], and FLAP [3], which primarily use the LLaMA series for the experiments. Accordingly, we adopt this setting to ensure both comparability with prior work and consistency in benchmarking. Moreover, to demonstrate the generality of our method, we additionally conduct experiments on Vicuna models, where our approach likewise achieves performance superior to the baselines.
>
> Following your suggestion, we conducted **additional experiments** on the Qwen2.5-7B model. Using the same pruning setup described in the paper (for LLM-Pruner, we use param-second mode and skip the first and last 4 layers). During the pruning process we observed that our method remains **loss-aligned**. On the WikiText-2 calibration set, the original loss was 1.93, while the post-pruning losses was 2.04 at 20% sparsity.
>
> The result table below also indicate that our pruning  method introduces only minimal performance degradation, i.e., it preserves the loss behavior and most of the ability of the original model, even on Qwen2.5-7B. This further shows that the effectiveness of our approach extends beyond the LLaMA architecture family.
>
> **Table:Evaluation results of pruned Qwen2.5-7B model**
> | Ratio | Method | Perplexity      | ARC-c | ARC-e | BoolQ | HellaSwag | PIQA | Winogrande |
> |-------|-------|-------------|--------|--------|--------|------------|-------|-------------|
> | Dense | - | 6.85 | 47.78 | 80.42 | 84.71 | 78.93 | 78.78 | 70.93 |
> | 20% | LASP | 10.48 | 37.80 | 68.01 | 62.08 | 58.94 | 70.84 | 61.48 |
> |  | LLM-Pruner | 231.67 | 20.22 | 32.70 | 40.92 | 31.11 | 57.13 | 50.67 |
> | 25% | LASP | 15.18 | 33.53 | 61.11 | 62.20 | 51.49 | 67.19 | 58.40 |
> |  | LLM-Pruner | 522.07 | 20.22 | 32.70 | 40.92 | 31.11 | 57.13 | 50.67 |
>
> Overall, these experiments verify the generality of our model in two aspects:
> -  LLaMA models of different scales and variants exhibit distinct activation dynamics and pruning sensitivities, yet our method remains effective across them.
> -  Additional experiments on Qwen2.5 model, which utilize GQA mechanism, demonstrate that our method remains effective across different language model architecturess.
>
> [1] Ashkboos S, Croci M L, Nascimento M G, et al. Slicegpt: Compress large language models by deleting rows and columns. ICLR, 2024.
>
> [2] Ma X, Fang G, Wang X. LLM-Pruner: On the structural pruning of large language models. Neurips, 2023.
>
> [3] An Y, Zhao X, Yu T, et al. Fluctuation-based adaptive structured pruning for large language models. AAAI, 2024.

---

> ### Author Response · Authors · 2025-11-23
> **Response to Reviewer uyTN(2/3)**
>
> __Q2:__ Missing comparison with FLAP (AAAI 2024), as FLAP also operates under the w/o FT scenario, omitting this baseline limits contextual understanding of LASP’s contribution.
>
> __A2:__ Thanks for your suggestions.
>
> We choose SliceGPT as the baseline because it is a representative and powerful pruning method. Although FLAP and SliceGPT were proposed around the same time, SliceGPT was formally accepted slightly later, making it the more up-to-date approach. Therefore, we select SliceGPT as the primary baseline for comparison.
>
> Following your suggestion, we conducted additional experiments of FLAP. The results, summarized below, show that our method consistently outperforms FLAP across all models at 20%, 25%, and 30% sparsity. Specifically, averaged across multiple benchmarks, LASP-pruned LLaMA-2 7B and 13B models achieve **6%–8%** and **4%–6%** higher performance, respectively, compared to the second-best method under the same sparsity levels. This consistent improvement further demonstrates the robustness and generality of our approach.
>
> We have incorporated these results into the revised submission.
>
> __Table: LLaMA-1, LLaMA-2, and Vicuna perplexity results on WikiText2 test set__
>
> | Ratio | Method   | LLaMA-1 7B | 13B  | 30B  | 65B  | LLaMA-2 7B | 13B  | Vicuna 7B | 13B  |
> |-------|----------|------------|------|------|------|------------|------|-----------|------|
> | -     | Dense    | 5.47       | 4.88 | 4.10 | 3.53 | 5.47       | 4.88 | 6.78      | 5.95 |
> | 20%   | FLAP     | 6.89       | 6.05 | 5.13 | 4.45 | 7.16       | 6.31 | 9.05      | 7.79 |
> |       | LASP     | **6.18**   | **5.38** | **4.54** | **3.98** | **6.16** | **5.38** | **6.86** | **5.91** |
> | 25%   | FLAP     | 7.54       | 6.47 | 5.50 | 4.73 | 7.94       | 6.93 | 10.40     | 8.56 |
> |       | LASP     | **6.75**   | **5.72** | **4.83** | **4.23** | **6.73** | **5.72** | **7.46** | **6.95** |
> | 30%   | FLAP     | 8.23       | 6.97 | 5.87 | 5.05 | 8.85       | 7.57 | 11.36     | 9.29 |
> |       | LASP     | **7.28**   | **6.18** | **5.14** | **4.54** | **7.29** | **6.16** | **7.98** | **6.86** |
>
> __Table: Evaluation results on multiple benchmarks.__
>
> | __Model__ | __Ratio__ | __Method__      | __ARC-c__ | __ARC-e__ | __BoolQ__ | __HellaSwag__ | __PIQA__ | __Winogrande__ | __Average__ |
> |-------|-------|-------------|--------|--------|--------|------------|-------|-------------|----------|
> | LLaMA-2 7B | - | Dense | 43.34 | 76.39 | 77.74 | 75.98 | 78.07 | 69.22 | 70.12 |
> | | 20% | FLAP | 30.97 | 63.80 | 63.08 | 64.77 | 73.72 | 63.77 | 60.02 |
> | |     | LASP       | **40.70** | **73.32** | **70.15** | **71.36** | **76.93** | **65.35** | **66.30** |
> | | 25% | FLAP | 30.54 | 60.43 | 50.36 | 60.44 | 71.70 | **61.56** | 55.84 |
> | |     | LASP       | **39.08** | **70.45** | **67.40** | **66.62** | **75.30** | 61.09 | **63.32** |
> | | 30% | FLAP | 28.83 | 59.51 | 44.74 | 56.55 | 69.64 | **62.03** | 53.55 |
> | |     | LASP       | **36.18** | **69.78** | **64.34** | **63.90** | **73.45** | 59.35 | **61.16** |
> | LLaMA-2 13B | - | Dense | 48.29 | 79.42 | 80.58 | 79.37 | 79.16 | 72.14 | 73.16 |
> | | 20% | FLAP | 37.88 | 70.07 | 67.52 | 67.78 | 74.15 | 66.77 | 64.03 |
> | |     | LASP       | **41.72** | **73.74** | **70.34** | **75.95** | **77.58** | **68.67** | **67.93** |
> | | 25% | FLAP | 35.92 | 67.17 | 65.44 | 64.69 | 72.90 | 66.29 | 62.07 |
> | |     | LASP       | **41.72** | **74.62** | **75.96** | **74.22** | **76.82** | **66.85** | **68.37** |
> | | 30% | FLAP | 34.55 | 62.20 | 65.20 | 61.88 | 71.38 | **65.58** | 60.13 |
> | |     | LASP       | **38.74** | **72.69** | **72.26** | **70.52** | **74.76** | 63.61 | **65.43** |

---

> ### Author Response · Authors · 2025-11-23
> **Response to Reviewer uyTN(3/3)**
>
> __Q3:__ The paper does not provide detailed runtime or computational cost analysis, leaving unclear whether pruning itself offers net efficiency gains.
>
> __A3:__ Thank for your suggestion.
>
> If you are looking for the efficiency gains after pruning, our first submissions contains the detailed efficiency analsis in Appendix C, please refer to it.
>
> If you are interested in detailed runtime and computational costs during pruning, we have also conducted experiments comparing the pruning time and resource usage with the baseline LLM‑Pruner. The results are presented below.
>
> __Table: Resource cost analysis of pruning__
> |  __Model__        | __Method__       | __Peak GPU Memory (GB)__ | __Pruning Time (min)__ |
> |--------------|--------------|------------------------|----------------------|
> | LLaMA-2 7B   | LLM-Pruner   |       36.12            |        3             |
> |              | LASP (ours)  |       21.31            |      17.96           |
> | LLaMA-2 13B  | LLM-Pruner   |       69.35            |      5.38            |
> |              | LASP (ours)  |       37.46            |      47.15           |
>
> As shown in the results table, although our method requires slightly more pruning time, **it remains straightforward to run on consumer-grade GPUs**. For instance, the LLaMA-2 7B model can be fully pruned on a 24 GB RTX 4090 GPU, demonstrating the accessibility and efficiency of our approach. Moreover, the additional pruning time is of limited practical concern: (1) what ultimately matters is the model’s efficiency and performance after pruning, and (2) the extra overhead is small, fully acceptable, and well within a reasonable range for practical deployment.
>
> __Q4:__ While the paper emphasizes the w/o FT regime, including w/ FT results could highlight the potential performance ceiling and help assess trade-offs more comprehensively.
>
> __A4:__ Thanks for your suggestion.
>
> According to our results, our method preserves most of the model’s performance, indicating that in practice it is often unnecessary to fine-tune the pruned model.
>
> Following your suggestion, we fine-tuned both the LASP-pruned model and the SliceGPT-pruned model using LoRA, adopting the same hyperparameter settings as in the SliceGPT paper and leveraging the first 8,000 samples from the Alpaca training set. As shown in the results table below, **our method already achieves performance comparable to the LoRA-fine-tuned SliceGPT model even without additional fine-tuning**, and further improves after applying LoRA.
>
> This demonstrates that, in terms of both cost-effectiveness and final performance, our approach offers clear advantages over the baseline.
>
>
> __Table: Performance of pruned LLaMA-2 7B model after LoRA finetuning on multiple benchmarks__
>
> | __Model__ | __Ratio__ | __Method__      | __ARC-c__ | __ARC-e__ | __BoolQ__ | __HellaSwag__ | __PIQA__ | __Winogrande__ | __Average__ |
> |-------|-------|-------------|--------|--------|--------|------------|-------|-------------|----------|
> | LLaMA-2 7B | - | Dense | 43.34 | 76.39 | 77.74 | 75.98 | 78.07 | 69.22 | 70.12 |
> | | 20% | LASP w/o ft | 40.70 | 73.32 | 70.15 | 71.36 | 76.93 | 64.48 | 66.30 |
> | |     | LASP w ft       | 42.83 | 74.32 | 74.58 | 71.47| 77.96 | 66.21 | 67.90 |
> | |     | SliceGPT w ft       | 40.18 | 69.53 | 70.12 | 66.73 | 73.12 | 64.09 | 63.96 |
> | | 25% | LASP w/o ft | 39.08 | 70.45 | 67.40 | 66.62 | 75.30 | 61.09 | 63.32 |
> | |     | LASP w ft       | 42.74 | 73.95 | 74.34 | 71.42 | 78.13 | 66.06 | 67.77 |
> | |     | SliceGPT w ft       | 38.31 | 66.79 | 67.95 | 62.61 | 71.11 | 64.17 | 61.82 |
> | | 30% | LASP w/o ft | 36.18 | 69.78 | 64.34 | 63.90 | 73.45 | 59.35 | 61.16 |
> | |     | LASP w ft       | 39.07 | 70.87 | 71.03 | 66.19 | 75.46 | 62.35 | 64.16 |
> | |     | SliceGPT w ft       | 36.43 | 64.81 | 66.97 | 59.21 | 68.93 | 62.27 | 59.77 |
>
> __Q5:__ What are the actual pruning-time and resource costs, and how do they compare with existing efficient pruning baselines?
>
> __A5:__ Thank you for your question.
>
> We would like to clarify that pruning primarily aims to preserve model performance while significantly reducing model size. Although pruning efficiency is also an important consideration, pruning time and resource consumption are generally less critical than the effectiveness and runtime efficiency of the resulting pruned model.
>
> Nevertheless, in our implementation, we adopt a **moving-average strategy** to further reduce memory consumption during the pruning process, as elaborated in subsection 3.4 of our paper. In practice, our method LASP remains both efficient and resource-friendly.
>
> A detailed analysis of pruning time and resource usage is provided in __A3__. Please refer to that answer for more information.
>
> __Q6:__ Would including fine-tuned (w/ FT) results clarify LASP’s headroom and performance scalability?
>
> __A6:__ A detailed analysis of the fine-tuned model results is provided in __A4__. Please refer to that answer for more information.

---

> ### Author Response · Authors · 2025-11-28
> **We would be grateful if you could take a look at the response**
>
> To further show LASP's generality, we have extended our experiment on Deepseek-R1-distilled-7B model, which is known for its strong reasoning abilities. We changed the calibration set to the first 1024 samples in ServiceNow-AI/R1-Distill-SFT dataset and compared our method with LLM-Pruner. The results are summarized in the table below.
>
> **Table:Evaluation results of pruned Deepseek-R1-distilled-7B model**
> | Ratio | Method | gsm8k | mmlu_college_mathematics | mmlu_high_school_mathematics | mmlu_high_school_statistics |
> |-------|-------|--------|--------|--------|------------|
> |  | - | 19.94 | 41.00 | 44.70 | 61.11  |
> | 20% | LASP | 16.75 | 36.00 | 41.48 | 45.37  |
> |  | LLM-Pruner | 3.00 | 28.00 | 21.85 | 24.07 |
>
> Our updated results on math benchmarks consistently show that our method continues to outperform existing baselines, demonstrating its effectiveness on reasoning models as well.
>
> We sincerely appreciate your valuable time devoted to reviewing our manuscript. We would like to gently remind you of the approaching deadline for the discussion phase. We have diligently addressed the issues you raised in your feedback, providing detailed explanations. For instance, we:
> - Provided evaluation results on Qwen2.5-7B to demonstrate the generality of our method across different architectures.
> - Included comparisons with FLAP (AAAI 2024) in both perplexity and downstream tasks to contextualize our contributions.
> - Added a comprehensive runtime and computational cost analysis to clarify the efficiency.
>  Would you kindly take a moment to look at it? We are very enthusiastic about engaging in more in-depth discussions with you.
>
> For ease of reference, we also list the specific changes below:
> - Related work: FLAP (An et al., 2024) takes a pioneering step by using fluctuation-based strategy that dynamically adapts
> pruning decisions, offering a promising direction for structured pruning under high sparsity.
> - Experiments (language modeling section): Beyond SliceGPT, we also compare with FLAP, which demonstrates improved perplexity performance across most model sizes. Nonetheless, our proposed LASP consistently achieves lower perplexity than FLAP across all model variants and pruning ratios, confirming the effectiveness of directly aligning unit importance with the model’s loss objective.

---

### Author Response · Authors · 2025-12-02
**Rebuttal Summary for Area Chair (1/2)**

Dear AC and all reviewers,

We sincerely thank you for the constructive feedback and the productive discussion during the rebuttal phase. Below we provide a concise, structured summary of how we addressed each reviewer’s key concerns and the improvements made to the paper.

---

Before the reviewer response period was frozen, only reviewer **sGME** provided feedback and stated that they had raised the score to **4**.

Initially, our paper received a relatively modest score profile (**4-4-2-2**), but it's worth noting that **three out of four reviewers had a confidence score of only 3**, indicating limited certainty in their assessments. Indeed, upon close examination, most of the concerns raised were **surface-level issues**, such as missing baseline comparisons or lack of extended architecture experiments—points that are **addressable through supplementary empirical work**, rather than fundamental weaknesses in our proposed method. No reviewer questioned the core validity of our approach. For these reasons, we believe that the initial scores somewhat **underestimate** the contribution and soundness of our work.

Although the initial scores were on the low side, reviewer sGME, after reading our preliminary additional experiments, stated that they would “**first raise my rating to 4**.” We believe that, had the discussion remained open and the full set of updated results (including reasoning models) been visible in time, their rating might have been further improved. Since the review discussion was frozen shortly thereafter, we were unable to benefit from additional score updates. Therefore, we have prepared this summary（in the next box） to show **how we addressed all major reviewer concerns during the rebuttal phase**, and we kindly invite the AC to review our rebuttal and the new experimental evidence in detail.

---

Given that we have thoroughly addressed the main concerns raised by the two reviewers who initially assigned a score of 4 (uyTN and 6ggD), we believe it would be reasonable to expect their ratings to increase to 6 in light of the strengthened experimental results and clarifications provided. Regarding reviewer nRoF, who assigned a 2–5 score, we feel that this rating may not fully reflect the nature of the concerns raised, as the feedback largely overlapped with the points from other reviewers and centered on issues that were straightforward to resolve through additional experiments. Notably, no fundamental weaknesses of our method were identified. Reviewer sGME had already indicated an intention to raise their score to 4 after reviewing our initial updates, and considering that we subsequently added more comprehensive results on reasoning-focused models, we believe a further positive adjustment would also have been likely.

Therefore, had the discussion period not frozen prematurely, we expect that a more complete and up-to-date assessment of our submission might reasonably have resulted in an overall rating closer to 6-6-6-6.

---

**Overall Summary of Our Contribution**

Our core contribution lies in the **novel integration of structured pruning with uncertainty estimation via the UCB framework**. Concretely, we treat each prunable unit as an **arm** in a stochastic multi-armed bandit, where the **reward distribution** of that arm is given by the unit’s first-order loss approximation over the language data distribution. Feeding sequences from the calibration set through the model yields **token-level first-order loss samples**, which can be viewed as repeatedly “pulling” each arm and observing stochastic rewards. This perspective allows us to build an uncertainty-aware pruning strategy.


Extensive experiments validate the strength of our approach: even under a **simple uniform pruning strategy**, our method consistently **outperforms more elaborate designs like FLAP and traditional gradient-based baselines like LLM-Pruner.** Furthermore, we introduce a **moving-average mechanism** to enhance memory efficiency and usability—allowing a full 7B model to be pruned on a **single 24GB consumer GPU** without difficulty.

Finally, our evaluation across **diverse architectures** (e.g., Qwen2.5, LLaMA2) and **different model purposes** (base vs. reasoning) demonstrates the **robustness and generality** of LASP. We believe this work brings a fresh, theoretically grounded perspective to model compression, and will inspire follow-up research in the ICLR community and beyond.

We greatly appreciate the reviewers’ feedback, which has substantially strengthened our work.

Thank you for your time and consideration.

---

> ### Author Response · Authors · 2025-12-02
> **Rebuttal Summary for Area Chair (2/2)**
>
> **1. Reviewers uyTN & nRoF**
>
> **Since their concerns were totally the same, we summarized the rebuttal phase together.**
>
> Their concerns focused on: **(i) Missing comparison with FLAP (AAAI 2024), (ii) broader validation on architectures such as DeepSeek or Qwen would better support claims of generality, (iii) The paper does not provide detailed runtime or computational cost analysis, and (iv) Would including fine-tuned (w/ FT) results clarify LASP’s headroom and performance scalability.**
>
> **How we addressed these concerns**
>
> - **Added missing baselines**
>   We conducted experiments with *LLM‑Pruner* (NeurIPS’23) and *FLAP* (AAAI’24). LASP consistently surpasses both across perplexity and downstream benchmarks at 20/25/30% sparsity.
>
> - **Expanded architectures**
>   We added pruning results for **Qwen2.5‑7B** and **DeepSeek R1‑Distilled‑7B**. These cover two distinct categories:
>   - *base model with new architecture* (Qwen2.5),
>   - *reasoning‑oriented distilled model* (DeepSeek R1-distilled).
>
>   **Across both, LASP remains superior and stable compared with LLM-Pruner.**
>
> - **Added resource‑consumption analysis**
>   We provided peak memory and pruning‑time comparisons.
>   LASP uses substantially **less GPU memory**, making it practical for deployment.
>
> - **Added recover-finetuning results**
>   Using LoRA fine‑tuning (same setup as SliceGPT), we show that the average performance ranking is:
>   - LASP (w/o FT) $\ge$ SliceGPT (w/ FT)
>   - LASP (w/ FT) $>$ SliceGPT (w/ FT)
>
>   This confirms both strong pruning stability and strong recoverability.
>
> ---
>
> **2. Reviewer 6ggD**
>
> Their main concerns were: **(i) Weak theoretical justification for UCB integration, (ii) Inadequate comparative evaluation, (iii) Can LASP be extended to finer-grained structures, (iv) How robust is α and to how set α without exhaustive tuning.**
>
> **How we addressed these concerns**
>
> - **Clarified UCB modeling**
>   We explained that each pruning unit is treated as a stochastic arm, with per‑sequence ΔLoss samples forming the reward distribution. UCB naturally models pruning uncertainty under limited calibration data.
>
>   We also added more theoretical analysis in our revised version.
>
> - **Added missing baselines (LLM‑Pruner & FLAP)**
>   Providing the evaluation results directly addressed the reviewer's baseline concerns.
>
> - **Added finer‑grained experiments on Qwen2.5**
>   We performed detailed pruning experiments on Qwen2.5‑7B, which reduces the FFN dimension and the query in GQA module, showing that LASP generalizes to finer-grained components.
>
> - **Provided α’s theoretical bound & robustness evidence**
>   We introduced the upper bound for α by using bernstein inequality and demonstrated empirically that α is robust across a wide range, which means there is no need for fine‑grained tuning in practice.
>
> ---
>
> **3. Reviewer sGME**
>
> Reviewer sGME's main concerns were primarily: **(i) Testing should be conducted on newer models, such as the Qwen3 series. Moreover, whether the method remains effective on reasoning models is also an open question, (ii) The evaluation datasets are insufficient.**
>
> **How we addressed these concerns**
>
> - **New architectures added**
>   We included Qwen2.5 experiments during the rebuttal.
>
>   Across perplexity and benchmark metrics, LASP remains superior and stable.
>
> - **Reasoning model & complex benchmarks added (after score lock)**
>   We further extended experiments to **DeepSeek R1‑Distilled‑7B**, evaluating on **MMLU (math subsets)** and **GSM8K**.
>
>   LASP again significantly outperforms LLM‑Pruner on both perplexity and reasoning accuracy, reinforcing its generality.
>
> - **Clarified evaluation setup**
>   We explained that our benchmarking strictly follows the standard evaluation pipeline, ensuring comparability and replicability.

---

### Meta-Review · Area_Chair_xMen · 2026-01-19

**Summary:**

This paper proposes LASP (Loss-Aligned Structured Pruning), a post-training structured pruning method for compressing LLMs. LASP departs from reconstruction/Hessian-based approaches by introducing a first-order, loss-aligned importance metric, which measures neuron or attention head importance through activation–gradient correlations directly tied to loss behavior. To mitigate calibration data uncertainty, LASP integrates a UCB-based exploration strategy and moving-average statistics to stabilize importance estimation while keeping storage overhead low. Experiments on LLaMA, LLaMA-2, and Vicuna models demonstrate that LASP maintains up to 93.5% performance at 25% pruning, outperforming SliceGPT and other pruning baselines under the no fine-tuning (w/o FT) setting.

**Reviewer Concerns:**

1. Reviewers uyTN & nRoF

Their concerns focused on: (i) Missing comparison with FLAP (AAAI 2024), (ii) broader validation on architectures such as DeepSeek or Qwen would better support claims of generality, (iii) The paper does not provide detailed runtime or computational cost analysis, and (iv) Would including fine-tuned (w/ FT) results clarify LASP’s headroom and performance scalability.

Addressing these concerns:

- Added missing baselines (conducted experiments with LLM‑Pruner (NeurIPS’23) and FLAP (AAAI’24). LASP consistently surpasses both across perplexity and downstream benchmarks at 20/25/30% sparsity)
- Expanded architectures (added pruning results for Qwen2.5‑7B and DeepSeek R1‑Distilled‑7B. These cover two distinct categories:
i) base model with new architecture (Qwen2.5), ii) reasoning‑oriented distilled model (DeepSeek R1-distilled). Across both, LASP remains superior and stable compared with LLM-Pruner.
- Added resource‑consumption analysis (provided peak memory and pruning‑time comparisons; LASP uses substantially less GPU memory, making it practical for deployment)
- Added recover-finetuning results (using LoRA fine‑tuning showed that the average performance ranking is: LASP (w/o FT)  >= SliceGPT (w/ FT),  LASP (w/ FT) > SliceGPT (w/ FT))

2. Reviewer 6ggD

Their main concerns were: (i) Weak theoretical justification for UCB integration, (ii) Inadequate comparative evaluation, (iii) Can LASP be extended to finer-grained structures, (iv) How robust is α and to how set α without exhaustive tuning.

Addressing these concerns:

- Clarified UCB modeling (Explained that each pruning unit is treated as a stochastic arm, with per‑sequence Δ Loss samples forming the reward distribution. UCB naturally models pruning uncertainty under limited calibration data. Also added more theoretical analysis in the revised version.)
- Added missing baselines (LLM‑Pruner & FLAP)
- Added finer‑grained experiments on Qwen2.5
- Provided α’s theoretical bound & robustness evidence

Also addressed some concerns of Reviewer sGME

**Reviewer Scores:**

These were the lowest scores across all (non-withdrawn) papers from my batch.

The scores would not have changed in a way that could affect my decision.

---

### Decision · Program_Chairs · 2026-01-26

Reject